# Evaluation of impacts of future climate change and water use scenarios on regional hydrology

**Seungwoo Chang[1], Wendy Graham[1, 2], Jeffrey Geurink[3], Nisai Wanakule[3], and Tirusew Asefa[3]**

[1]Water Institute, University of Florida, 570 Weil Hall, PO Box 116601, Gainesville, FL 32611, USA

[2] Department of Agricultural and Biological Engineering, University of Florida, 570 Weil Hall, PO Box 116601, Gainesville, FL 32611, USA

[3] Tampa Bay Water, 2575 Enterprise Rd, Clearwater, FL 33763-1102, USA

Corresponding author: S. Chang (swjason@ufl.edu)

**Abstract**

General circulation models (GCMs) have been widely used to simulate current and future climate at the global scale. However, the development of frameworks to apply GCMs to assess potential climate change impacts on regional hydrologic systems, the ability to meet future water demand, and compliance with water resource regulations is more recent. In this study eight GCMs were bias-corrected and downscaled using the Bias Correction and Stochastic Analog (BCSA) downscaling method and then used, together with three $ET_0$ methods, and eight different water use scenarios to drive an integrated hydrologic model previously developed for the Tampa Bay region in west central Florida. Variance-based sensitivity analysis showed that changes in projected streamflow were very sensitive to GCM selection, but relatively insensitive $ET_0$ method or water use scenario. Changes in projections of groundwater level were sensitive to both GCM and water use scenario, but relatively insensitive to $ET_0$ method. Five of eight GCMs projected a decrease in streamflow and groundwater availability in the future regardless of water use scenario or ET method. For the business as usual water use scenario all 8 GCMs indicated that, even with active water conservation programs, increases in public water demand projected for 2045 could not be met from ground and surface water supplies while achieving current

groundwater level and surface water flow regulations. With adoption of 40% wastewater reuse for public supply and active conservation 4 of the 8 GCMs indicate that 2045 public water demand could be met while achieving current environmental regulations; however, drier climates would require a switch from groundwater to surface water use. These results indicate a high probability of a reduction in future freshwater supply in the Tampa Bay region if environmental regulations intended to protect current aquatic ecosystems do not adapt to the changing climate. Broad interpretation of the results of this study may be limited by the fact that all future water use scenarios assumed that increases in water demand would be the result of intensification of water use on existing agricultural, industrial and urban lands. Future work should evaluate the impacts of a range of potential land use change scenarios, with associated water use change projections, over a larger number of GCMs.

## 1. Introduction

The Intergovernmental Panel on Climate Change (IPCC) along with many other studies have indicated that climate change is likely to alter both the global hydrologic cycle and regional hydrologic cycles (Aalst et al., 2014; Déry et al., 2009; Georgakakos et al., 2014; Hawkins et al., 2014; Milliman et al., 2008). These studies have indicated that climate change is likely to increase the frequency of droughts, as well as the magnitude of floods in many regions (Diffenbaugh and Field, 2013; Georgakakos et al., 2014; Walsh et al., 2014). It is necessary to investigate future climate change and its potential impacts on the natural environment in order to reduce risks and increase resilience for future water resources planning and management (Vano and Lettenmaier, 2013).

General Circulation Models (GCMs) and hydrologic models have been widely used to evaluate future climate change and its impact on regional hydrologic cycles (Boé et al., 2007; Maurer and Hidalgo, 2008). However, there are a variety of barriers to direct use of GCMs to drive regional hydrologic models. For example, the current generation of GCMs contain biases that prevent accurate reproduction of historic hydrological conditions when used to drive hydrologic models (Giorgi and Mearns, 2002; Wood et al., 2002). In addition, the coarse resolution of GCMs prevents direct use of their results with regional hydrologic models that require higher resolution climate variables (Solomon et al., 2007). Many bias correction methods and downscaling methods have been developed and evaluated to overcome these limitations

(Chen et al., 2013; Ghosh and Mujumdar, 2008; Hwang and Graham, 2013; Langousis et al.,
2015; Muerth et al., 2013; Quintana Seguí et al., 2010; Stoll et al., 2011; Zhang and
Georgakakos, 2012).  Although these bias correction and downscaling methods do not correct
problems with large scale synoptic forcing, and are not particularly good at reproducing extreme
floods or droughts in the retrospective period, previous research has shown that they are able to
simulate broad features of the climate system and are useful for characterizing plausible
projections of possible futures (Kundzewicz et al, 2008, 2009). Furthermore, previous work in
the study region has shown that hydrologic models driven by bias-corrected downscaled
retrospective GCM output adequately reproduce retrospective  high stream flows (e.g. 7Q2 and
7Q10), as well as the long term mean and standard deviation of monthly flows (Hwang and
Graham, 2014).

In addition to studies that focus on climate impacts on the hydrological cycle, it is also

necessary to evaluate the effects of direct human behavior (Haddeland et al., 2014; Wang and
Hejazi, 2011). Human activities such as agricultural production, irrigation (Gupta et al., 2015),
municipal pumping (Patterson et al., 2013), deforestation, and urban development alter regional
hydrologic behavior (Siriwardena et al., 2006). For robust water resources management and
planning better understanding of the influence and relative importance of climate change and
human-induced change on hydrology and water resources is essential (Chang et al., 2016; Ma et
al., 2008; Tan & Gan, 2015; Ye et al., 2013; Zheng et al., 2009).

The relative contributions of climate change and human activities to hydrologic responses

have been evaluated using GCM data to drive hydrologic models with plausible future
anthropogenic scenarios (Liu et al., 2013; Maurer et al., 2010; Wood et al., 2002). Murray et al.
(2012) used the Land-surface Processes and eXchanges (LPX) dynamic global vegetation model
and the WaterGAP hydrological model to evaluate the impacts of climate change and socio-
economic change on global hydrologic response for the 2070 – 2099 time period. They found
that climate change and population growth increased water stress in many regions, and change in
runoff was most highly correlated with precipitation change in large global catchments. Harding
et al. (2012) applied downscaled outputs of 16 GCMs with the VIC model to investigate the
future change in streamflow for the Colorado river basin. They suggested that impact analyses
relying on only a few scenarios were unacceptably influenced by the choice of GCM projections.
For studies using GCMs to project future hydrologic responses, uncertainties resulting
from the choice of GCM, RCP (Representative Concentration Pathways) trajectory, and
reference evapotranspiration ($ET_0$) estimation methods are all significant, and it is important to
quantify the relative uncertainties of these factors (Chang et al., 2016; Hawkins & Sutton, 2009,
2010; Kingston et al., 2009; Koedyk & Kingston, 2016; McAfee, 2013; Thompson et al., 2014;
W. Wang et al., 2015). Furthermore, the effects of climate change on groundwater levels have
not explored as extensively as the effects of climate change on surface water flows (Green et al.,
2011; Kløve et al., 2014). Kløve et al. (2014) suggested that the uncertainties of groundwater
projections attributed to climate models, downscaling techniques, emission scenarios, land use
changes and social economic development should be evaluated.
This study evaluated the future projections of regional hydrologic response using eight
GCMs, three $ET_0$ estimation methods, and eight human water use scenarios to drive a calibrated
regional hydrologic model developed for the Tampa Bay region. A comprehensive evaluation of
the relative sensitivity of projections of regional hydrologic response to the choice of GCM, $ET_0$
estimation method, and human water use scenario was conducted. Statistical analyses were
performed to determine whether differences in streamflow and groundwater level between
retrospective hydrologic and projected future climate were statistically significant given these
underlying prediction uncertainties. The ability to satisfy projected increases in future water
demand while meeting current groundwater level and surface water flow regulations was
evaluated over the suite of GCM and water management scenarios.
**2. Materials and Methods**
2.1 Study Region
Tampa Bay Water operates a diverse regional water supply system comprised of a
desalination plant, well fields that extract water from the Floridan Aquifer, and surface water that
is extracted from the Hillsborough and Alafia Rivers (https://tampabaywater.org/water-supply-
sources-tampa-bay-region ). The fresh groundwater system in the region is composed of two
aquifer systems, a thin surficial aquifer and the thick and highly productive carbonate rocks of
the Floridan aquifer system (Tihansky & Knochenmus, 2001). Dynamic interacting surface-
water and groundwater systems (in which groundwater from in the aquifer used for agricultural
irrigation and public water supply also feeds the surface springs and rivers) characterize the
region and must be considered in the management of water resources (Tihansky, 1999). For
example the SWFWMD regulates groundwater pumping for water supply to maintain
groundwater levels that promote environmental protection of lakes and wetlands near well-fields.
Similarly they regulate the daily volume of flow permitted for extraction from rivers based on
maintaining sufficient in-stream flows and spring flows to protect aquatic ecosystems.

This study focused on the Integrated Northern Tampa Bay (INTB) model domain

(Geurink and Basso, 2013; Hwang and Graham, 2014). Figure 1 shows the INTB model domain,
model sub-basins, locations of agricultural, industrial and public water supply wells, two
streamflow locations where water is withdrawn for public supply, and three monitoring wells
near Tampa Bay Water's consolidated well fields that are used to evaluate compliance with
groundwater level regulations. The INTB region land use currently consists of grass/pasture (25
%), urban (22 %), forested (15 %), mining/other (7 %), agriculture/irrigated land (6 %), open
water (4 %), and wetlands (21 %).

2.2 The Integrated Northern Tampa Bay Model

Tampa Bay Water and the Southwest Florida Water Management District (SWFWMD)

developed the Integrated Hydrologic Model (IHM) simulation engine which integrates the EPA
Hydrologic Simulation Program-Fortran (Bicknell et al., 2005) for surface water modeling with
the U.S. Geological Survey (USGS) MODFLOW96 (Harbaugh and McDonald, 1996) for
groundwater modeling. The IHM simulates the dynamic interaction of surface water and
groundwater systems within the INTB region including all processes which affect flow and water
levels in uplands, within the unsaturated soil, and within wetlands, rivers and aquifers. In
addition, the INTB model can account for variability in climate and anthropogenic stresses such
as land use change, groundwater pumping, and diversions to/from rivers, lakes, and wetlands.

Tampa Bay Water and the SWFWMD calibrated model parameters to simulate

streamflows, groundwater levels, and wetland hydroperiods in the INTB model region. The
INTB model was calibrated from 1989 to 1998 and verified from 1999 to 2006 (Geurink and
Basso, 2013). Precipitation data for calibrating and validating the model were obtained from 302
point gages maintained by National Oceanic and Atmospheric Administration (NOAA), the
SWFWMD, and Tampa Bay Water in the model region. Maximum and minimum daily
temperature were obtained from six NOAA stations within the INTB region and used to estimate
$ET_0$ using the Hargreaves method. Over the calibration and validation period (1989 to 2006)
average annual precipitation input to the model was 1308 mm/year and average annual actual
evapotranspiration estimated by the model was 940 mm/year, resulting in net available water
(precipitation-actual evapotranspiration) of 368 mm/yr. During this period surface discharge
from the domain was 272 mm/year (74 % of net available water), groundwater pumping was 69
mm/year (19 %), surface water diversions for water supply were 10 mm/year (3 %), and
irrigation applied within the domain was 18 mm/year (5 %). More details about the processes
and results of model calibration and validation are described in Geurink and Basso (2013).
Streamflow predictions at two United States Geological Survey (USGS) gauging
stations, the Hillsborough river (USGS ID: 02303330) and Alafia river (USGS ID: 02301500),
were used in this study to evaluate retrospective and future IHM streamflow predictions and
quantities of surface water available for public supply. Three Tampa Bay Water monitoring wells
(NWH-RMP-08s, CBR-SERW-s, and STK-STARKEY-20s) were used to evaluate retrospective
and future groundwater level predictions and compliance with environmental regulations
intended to protect nearby wetlands from dewatering as a result of consolidated well field
pumping.
2.3 Climate Data
Forcing data from Phase 2 of the North American Land Data Assimilation System
(NLDAS-2) from 1982 to 2005 were used as historical reference climate data for bias correction.
Hourly precipitation, air temperature, solar radiation (surface downward longwave radiation and
surface downward shortwave radiation), surface pressure and average wind speed were obtained
from the NLDAS-2 archive and aggregated to the daily scale at a 1/8th-degree grid spacing over
the Tampa Bay region.
For retrospective and future climate data, the Coupled Model Intercomparison Project 5
(CMIP5) General Circulation Models (GCMs) data set for the 1982-2005 period was used for the
retrospective period and 2030-2060 (Future 1) and 2070-2100 (Future 2) were used as future
periods. Gridded daily precipitation, air temperature, solar radiation, surface pressure, and
average wind speed were obtained for eight GCMs listed in Table 1. These GCMs were chosen
because they spanned the range of cool to warm bias and wet to dry bias exhibited by 41 CMIP5
GCMs for the southeastern United States (Rupp, 2016), and they had daily values available for
all the parameters needed to estimate Penman-Monteith reference evapotranspiration. Mean
changes in precipitation projected by these GCMs ranged from -68 mm/year to 293 mm/year
over the 2030-2060 period, and from 154 mm/year to 400 mm/year over the 2070-2100 period.
Mean changes in $ET_0$ ranged from 24 mm/year to 137 mm/year over the 2030-2060 period and
from 122 mm/year to 351 mm/year over the 2070-2100 period. Mean changes in $P-ET_0$ ranged
from -162 mm/year to 220 mm/year over the 2030-2060 period and from -420 mm/year to 159
mm/year over the 2070-2100 period (Table 1).
Chang et al. (2016) evaluated projected changes in $P - ET_0$ over the continental USA
using nine GCMs, ten $ET_0$ estimation methods, and three RCP scenarios. They showed that the
first order sensitivities of water deficit projections ($P-ET_0$) over the Southeast USA were much
higher to choice of GCM and $ET_0$ estimation method than to choice of RCP. First order
sensitivities of water deficit projections to RCP scenarios were negligible (<0.01) for the 2030-
2060 time period, and averaged 0.2 for the 2070-2100 time period. Therefore for computational
efficiency, and to evaluate the influence of the most extreme carbon dioxide forcing on the
hydrologic projections, only the RCP 8.5 scenario data was utilized for the future analyses in this
study.
2.4 BCSA Downscaling Method
The BCSA downscaling method, developed by Hwang and Graham (2013), was used in
this study. Hwang & Graham (2014) showed that BCSA demonstrated  better performance than
other statistical downscaling methods (i.e, BCSD (Maurer et al, 2012) or SDBC (Abatzoglou and
Brown, 2012)) in reproducing spatiotemporal statistics of both precipitation and daily streamflow
in the Tampa Bay region. In particular, the INTB model, when driven by GCMs downscaled
using the BCSA method, accurately reproduced frequencies of extreme high and extreme low
retrospective streamflows as well as 7Q2 and 7Q10 retrospective streamflows in the Tampa Bay
region.
The BCSA method preserves both the cumulative frequency distribution of observed
daily precipitation as well as the spatial autocorrelation structure of observed daily precipitation
fields. BCSA downscaling consists of two separate steps for bias-correction and stochastic
analog spatial downscaling. In the first step, a cumulative distribution function (CDF) mapping
approach ( Block et al., 2009; Hwang et al., 2013, 2014; Hwang & Graham, 2014; Ines &
Hansen, 2006; Teutschbein & Seibert, 2012) is used to reduce the biases in raw GCM output at
the GCM scale. In this study, NLDAS-2 P and $ET_0$ were aggregated up to the GCM scale and P
and $ET_0$ from the raw GCMs were bias corrected at the GCM scale using the sequential
univariate CDF mapping method (Chang, 2017). NLDAS-2 was selected for bias correction
because it includes all the parameters needed to estimate Penman-Monteith reference
evapotranspiration. Comparison of the gridded NLDAS-2 data to the precipitation and
temperature observations from the weather stations used to calibrate the INTB model showed
that the NLDAS-2 data reproduced observed long term monthly means values with biases that
ranged from 4 to 12 mm for daily precipitation and 1 to 2°C for daily temperature. Correlations
among daily values ranged from 0.75 to 0.87 for rainfall and 0.75 to 0.98 for temperature. The
second step in the BCSA method is stochastic analog (SA) spatial downscaling (Hwang &
Graham, 2013, 2014) for P. In this method, a synthetic downscaled precipitation field is
produced which preserves the GCM-scale daily precipitation amount and the month-specific
local-scale spatial correlation structure. For more details on the BCSA method, see ( Hwang &
Graham, 2013, 2014). $ET_0$ was not downscaled in this study because observed spatial variability
of $ET_0$ over the INTB region is very small, and the spatial correlation is large compared to P
(Chang, 2017).
2.5 Reference Evapotranspiration Estimation Methods
The Chang et al. (2016) study referenced above found that the projected changes in P –
$ET_0$ were sensitive to both the choice of GCM and the choice of $ET_0$ method, and that for the
Southeast USA the choice of GCM and $ET_0$ method had approximately equal influence on
changes in future P – $ET_0$ throughout most of the year. However, they noted that not all ten $ET_0$
methods were equally appropriate for use in all US regions, and that regional studies should use
methods for which retrospective predictions of $ET_0$ are generally consistent with historic
observations. Several of the $ET_0$ methods used by Chang et al. (2016) were found to produce
unreasonably high or low historic $ET_0$ estimates for the study region using retrospective and
observation data. Therefore in this study three $ET_0$ estimation methods that are widely used in the
Southeast USA, produced retrospective predictions that were consistent with observations, and
showed a range of wet to fairly dry projections of future P-$ET_0$ (Chang et al, 2016) were
included in the analysis. These methods include a temperature-based method (Hargreaves;
Hargreaves and Allen, 2003), a radiation-based method (Priestley-Taylor; Allen et al., 1998), and
a combination method (Penman-Monteith; Allen et al., 1998). All hourly climate variables
described above were aggregated to the daily scale and used to calculate daily $ET_0$ using these
three methods.

2.6 Retrospective Simulations

Water-use in the  study region is comprised of five categories; 1) public supply, 2)

agricultural, 3) industrial/commercial, 4) mining, and 5) recreational (e.g. golf course irrigation)
(Geurink and Basso, 2013).  Groundwater sources are used for agricultural,
industrial/commercial, mining and recreational water supplies. Public water supply is provided
by a combination of groundwater, surface water (Hillsborough and Alafia Rivers), and a 25
MGD desalinization plant operated by Tampa Bay Water. The SWFWMD regulates all
groundwater pumping and surface water extraction in the study region to protect natural aquatic
ecosystems and prevent saltwater intrusion. Over the 1989-2006 calibration-verification period
groundwater extractions from the INTB model domain averaged 36 mm/yr for public water
supply, 18mm/yr for agricultural irrigation, 9 mm/year for  industrial/commercial uses, 6
mm/year for mining, and 3 mm/year for recreational uses (Geurink and Basso, 2013).

Public Water Supply: Tampa Bay Water has a consolidated permit for its eleven

wellfields (the Consolidated Wellfields, hereafter referred to as the CWF). The CWFs are
operated as an interconnected system with a combined maximum permitted pumping rate of 90
MGD (13 mm/yr over the INTB region). Individual well pumping rates are optimized to
maintain minimum groundwater levels near sensitive wetlands to meet regulatory requirements
intended to prevent ecological harm. The three monitoring wells evaluated in this study are
located near wetlands adjacent to the CWFs (Fig. 1). From 1992-2008 Tampa Bay Water's total
water demand averaged ranged from 150-200 MGD. Groundwater is Tampa Bay Water's most
inexpensive source for public water supply, therefore for the retrospective simulations the CWFs
were assumed to withdraw groundwater continuously at the 90 MGD maximum permitted rate.
For the retrospective simulations groundwater extraction for other public water supply (outside
of Tampa Bay Water's CWF), industrial/commercial and mining uses were assumed occur
continuously at the average pumping rates between years 1989 to 2006 cited above.
Maximum available surface water available to Tampa Bay Water for public supply was
calculated on a daily basis from retrospective streamflow predictions for both the Hillsborough
River and the Alafia River according to site-specific regulations set to maintain sufficient in-
stream flows and spring flows to protect aquatic ecosystems. Diversion rates for pumping from
the Hillsborough river reservoir by the City of Tampa and from the Tampa Bypass Canal by
SWFWMD were set at the historical average daily rate spanning 2003 to 2009 for all
retrospective simulations. All other diversion rates were set to zero including the Withlacoochee-
Hillsborough overflow. These diversion locations are located either downstream or outside of the
watersheds contributing to the surface water gages, and outside the zone of influence of the
monitoring wells evaluated in this study so these assumptions do not impact on the results (Fig.

1).

Agricultural Irrigation Demand: The AFSIRS (Agricultural Field-Scale Irrigation

Requirements Simulation) model (Jacobs and Dukes, 2007; Smajstrla, 1990) was used to
estimate climate-driven irrigation demand for the retrospective period. The AFSIRS model tracks
the water budget in the crop root zone including inputs from rain and irrigation, and outputs from
the root zone by drainage and evapotranspiration. The AFSIRS model defines the water storage
capacity in the crop root zone as the product of the water-holding capacity of the soil (estimated
by the difference between field capacity and wilting point) and the depth of the effective root
zone for the crop being grown. Crop evapotranspiration (ETc) is estimated from the product of
potential evapotranspiration ($ET_0$) and crop water use coefficients. The AFSIRS model
subdivides the crop root zone into irrigated and non-irrigated zones and maintains separate water
budgets for both zones in order to simulate different types of irrigation systems, such as surface
irrigation and subsurface irrigation.

The AFSIRS was used as a basis to estimate irrigation demand for the retrospective

period using CMIP5 bias-corrected downscaled daily P and bias-corrected $ET_0$ (using the three
$ET_0$ methods discussed above) and the land use from the calibrated INTB model. Crop
coefficients ($K_c$) for estimating $ET_c$ were obtained from the calibrated INTB model database
(Geurink and Basso, 2013) for all vegetative covers except row crops. The crop coefficient for
row crops was estimated by the superposition of crop coefficients for tomato and strawberry
(Dukes et al., 2012), the two dominant row crops in the region. The relative proportion of these
two crops constituting the row crop land use were calculated based on water usage records for
the region for 2011 (Jackson and Albritton, 2013). The root zone depth, field capacity, wilting
point and other information needed for the AFSIRS model were taken from the calibrated INTB
model database. Groundwater pumping required to satisfy the AFSIRS estimated irrigation
assumed 85% irrigation efficiency based on Irmak et al. (2011) and Jacobs & Dukes (2007), i.e.,
$$agricultural\ pumping = irrigation\ demand \times \frac{100\ \%}{85\ \%} \qquad (3)$$
It should be noted that the AFSIRS model does not predict water demand for bed
preparation, freeze protection, crop cooling requirements, or maintenance of irrigation systems.
Thus the irrigation demand estimated for the retrospective period only includes crop water
demand for evapotranspiration.
Boundary Conditions: Lateral boundary conditions are required for aquifers in the model
region. A repeating annual cycle of daily General Head Boundary (GHB) time series for the
retrospective and future periods IHM simulations was derived using the daily average of the
historical daily GHB time series spanning 2000 to 2006. More details about the water
withdrawals such as groundwater pumping, agricultural irrigation, CWFs, diversions and
boundary conditions during the calibration-verification period are described in Geurink and
Basso (2013).
2.7 Future Water Use Scenarios
In addition to warming temperatures and reduced precipitation due to climate change,
increases in water withdrawal for agriculture and other human uses are potentially significant
causes of declining river flow and groundwater levels (Alcamo et al., 2003; Vorosmarty et al.,
2000). To assess the relative importance of climate change versus anthropogenic impact on the
hydrologic system, ability to meet future water demand, and compliance with water resource
regulations in the study region, eight future water use scenarios were developed (Table 2). These
scenarios were based on discussions with Tampa Bay Water staff, projected increases in public
water demand (Tampa Bay Water Water Demand Management Plan Final Report, 2013),
projected changes in agricultural land use and agricultural irrigation demand (Florida Statewide
Agricultural Irrigation Demand Report, 2017), potential agricultural adaption behaviors, and
potential changes in groundwater regulations. For naming simplicity in the future scenarios
agricultural and recreational water use categories are combined as agricultural demand and
public supply, industrial/commercial and mining are combined as urban demand. The eight water
use scenarios included: 1) No groundwater pumping for agriculture or urban demand, 2) No
urban groundwater pumping, 3) No agricultural groundwater pumping, 4) Agricultural adaption
(increased irrigation efficiency and/or use of reclaimed water), 5) Business as usual, 6) Increased
agricultural demand, 7) Relaxed regulatory requirements for CWF pumping (increased CWF
pumping), and 8) Relaxed regulatory requirements for all urban groundwater pumping (increased
all urban pumping). Details regarding each of these water use scenarios are provided below.

The business as usual scenario (scenario 5 in the Table 1) assumed no change in

groundwater regulations. Thus the CWF pumping remained at the maximum permitted 90 MGD
and all other urban pumping (industrial/commercial, mining and other public water supply)
remained at the average pumping rates used in the retrospective simulations.  In this case all
projected increases in future public water demand must be met by increased surface water
extraction (if available), increased conservation, increased wastewater reuse, or desalination
capacity. For the business as usual scenario agricultural irrigation demand was estimated using
AFSIRS model and assuming 85% irrigation efficiency, as in the retrospective period
simulations. However the P and $ET_0$ used in the AFSIRS model were taking from the bias
corrected downscaled future GCM projections for both future 1 (2030-2060) and future 2 (2070-

2100).

To more clearly separate the impact of human water use versus climate change on the

hydrologic system, three extreme groundwater use reduction scenarios were developed. The no
agricultural or urban pumping scenario (scenario 1) assumed that there was no groundwater
pumping at all in the region. For this scenario agricultural and recreational pumping (and the
associated irrigation of the land surface) as well as all urban pumping (including CWF, other
public water supply and industrial/mining) were set to zero. For the no urban pumping scenario
(scenario 2) all urban pumping including CWF, other public water supplies, industrial/mining
was set to zero, however agricultural pumping was assumed to be the same as the business as
usual scenario. For the no agricultural pumping scenario (scenario 3) agricultural and
recreational pumping were set to zero, however all urban pumping was assumed equal to the
business as usual scenario.
The agricultural adaption scenario (scenario 4) assumed that increased irrigation
efficiency and/or increased use of reclaimed water reduced groundwater pumping for agricultural
and recreational irrigation by 40 MGD over climate driven demand (6 mm/year, ~25%).  All
urban pumping was assumed to be the same as the business as usual scenario. The increased
agricultural demand scenario (scenario 6) assumed that irrigation demand increased by 40 MGD
over climate driven demand (6 mm/year, ~25%) due to more intensive farming on existing
agricultural lands (Florida Statewide Agricultural Irrigation Demand Report, 2017) and that all
urban pumping was the same as the business as usual scenario. The relaxed regulatory
requirements for CWF pumping (scenario 7) assumed an increase of CWF pumping up to 130
MGD (19 mm/year, ~44%) from the current 90 MGD (13 mm/year) to help meet increased
public water demand, and that agricultural and recreational pumping followed the business as
usual scenario. The relaxed regulatory requirements for all urban pumping (scenario 8) assumed
all urban pumping, including CWF pumping, other public water supply, industrial and mining,
increased by 44 %, (i.e. the same percentage increase as the CWF pumping for scenario 7) and
that agricultural and recreational pumping followed the business as usual scenario. These water
use scenarios consist of projected agricultural and urban groundwater pumping volumes that
represent from 0 % to 27 % of historic P-ET$_0$.
It should be noted that land use change was not considered in this study. This assumption
is consistent with a regional planning strategy that promotes agricultural and urban
intensification on existing lands, along with protection of existing conservation lands, wetlands
and water supplies (Barnett et al., 2007). This assumption is also consistent with the Florida
Statewide Agricultural Irrigation Demand Report (2017) that projects a 2% decline in
agricultural land area between 2015-2040, but an 8.5% increase in agricultural water use as a net
result of agricultural intensification and increased conservation. Future work will build on this
study to evaluate land use change scenarios.
2.8 Statistical Analysis
Variance-based sensitivity analysis is a global sensitivity analysis (GSA) method (Saltelli
et al., 2008, 2010) used to apportion the total model output variance simultaneously onto all the
varying input factors, and thus is preferred over the local, one factor at a time, sensitivity
analyses (Homma and Saltelli, 1996; Saltelli, 1999). In this research the sensitivity of projected
changes between future and retrospective mean monthly streamflow and groundwater levels was
evaluated using the variance-based GSA method described in Chang et al. (2016).
Using the variance-based GSA method the variance-based first order effect is expressed
as:
$$V_{X_i}\left(E_{X_{\sim i}}(Y|X_i)\right) \tag{1}$$
Where V is the scalar model output (i.e., change in mean monthly streamflow or
groundwater level), and $X_i$ are the factors causing variation in the model output ( i.e. choice of
GCM, $ET_0$ method, water use scenario). The expectation operator $E_{X_{\sim i}}(Y|X_i)$ indicates that the
mean of Y is taken over all possible values of $X$ except $X_i$ (i.e., $X_{\sim i}$ ) while keeping $X_i$ fixed. The
variance, $V_{X_i}$ , is then taken of this quantity over all possible values of $X_i$ . The first-order
sensitivity coefficient is
$$S_i = \frac{V_{X_i}(E_{X_{\sim i}}(Y|X))}{V(Y)} \tag{2}$$
where $V(Y)$ the total variance of Y over all $X_i$. $S_i$ is a normalized index varying between
0 and 1, because $V_{X_i}\left(E_{X_{\sim i}}(Y|X_i)\right)$ varies between 0 and $V(Y)$ according to the identity (Mood et
al., 1974):
$$V_{X_i}\left(E_{X_{\sim i}}(Y|X_i)\right) + E_{X_i}\left(V_{X_{\sim i}}(Y|X_i)\right) = V(Y) \tag{3}$$
The first-order sensitivities of future changes in mean seasonal streamflow and
groundwater level to the choice of GCM, $ET_0$ estimation method, and water use scenario were
calculated over the full ensemble of 8 GCMs, 3 $ET_0$ methods and 8 water use scenarios (192
samples) for each future period in order to evaluate the relative contributions of each of these
factors on the variation among projections of future changes.
In addition to variance-based GSA, differences in future changes of mean projected
streamflow and groundwater level across GCMs and across future water use scenarios were
evaluated for statistical significance using Tukey's HSD (honest significant difference) test
(Zieyel, 1988) that is a single-step multiple statistical test (pairwise comparison). The two-
sample t-test was used to test for significant differences between mean projected streamflow and
groundwater levels resulting from future climate/water use scenarios and mean retrospective
streamflow and groundwater level using the business as usual water use scenario.
**3 Results and Discussion**
3.1 Global Sensitivity Analysis of Projected Changes
The variance-based global sensitivity analysis was conducted for both the wet season
(June – September) and the dry season (October – May) to evaluate the relative variation of
projected changes in hydrologic response attributed to the choice of GCM, choice of water use
scenario, and choice of $ET_0$ method. Tables 3 and 4 show the first order sensitivity indices of
changes in future streamflow and groundwater level (defined as future average seasonal
streamflow – retrospective average seasonal streamflow and future average seasonal
groundwater level – retrospective average seasonal groundwater level, respectively).
Change in streamflow was much more sensitive to choice of GCM than to choice of $ET_0$
method or water use scenario for all river gages, both seasons, and both future periods (Table 3).
For example, 94.4% of the variance of the change in wet season Hillsborough river streamflow
in Future 1 period (2030-2060) is attributed to differences among GCMs, 0.2% of the variance is
attributed to differences among $ET_0$ method, and 1.6% of the variance is caused by water use
scenario, respectively (top row Table 3). Similarly, projected changes in groundwater level were
generally more sensitive to the choice of GCM for all monitoring wells and both seasons.
However, unlike the projected changes in streamflow, changes in groundwater level were also
quite sensitive to the choice of water use scenario (Table 4). The higher sensitivity of
groundwater level to groundwater pumping is expected since the monitoring wells are
intentionally located near the consolidated wellfields (locations of major groundwater pumping)
to detect and mitigate localized impacts of water supply pumping on nearby wetlands. On the
other hand, the stream gages are located further from the consolidated well fields and accumulate
flow from a large area of the model domain. The first order sensitivity index of groundwater
level to water use scenario decreased in future period 2 (2070-2100) over future period 1 (2030-
2060), due to the increased variability of GCM precipitation projections in future 2 (2070-2100)
versus future 1 (2030-2060).

As mentioned previously Chang et al. (2016) evaluated projected changes in $P - ET_0$ over

the continental USA using nine GCMs, ten $ET_0$ estimation methods, and three RCP scenarios
and found that for the Southeast USA the choice of GCM and $ET_0$ method had approximately
equal influence on changes in future $P - ET_0$ throughout most of the year. Because this study
eliminated several $ET_0$ estimation methods that produced unreasonably high and low historic $ET_0$
estimates for the study region using the NLDAS-2 data, the first order sensitivity index for $ET_0$ is
significantly lower in this study than in their results. It should be noted that these results do not
indicate that the choice of reference ET estimation method does not affect the change in
streamflow or groundwater, only that the choice of reference ET estimation method is much less
influential than the choice of GCM or choice of water use scenario.

3.2 Projections of Streamflow

The INTB was run to compare retrospective hydrologic response to historical

observations and model predictions generated with the calibrated model using NLDAS-2 data, as
well as to future hydrologic response as a result of alternative GCMs, $ET_0$ methods and water use
scenarios. Figure 2 shows observed, NLDAS-2 and retrospective mean monthly streamflow for
the Hillsborough river (Fig. 2a) and Alafia river (Fig. 2b), as well as future mean monthly
streamflow in future 1 (2030-2060) and future 2 (2070-2100) for the business as usual water use
scenario using the Hargreaves $ET_0$ method originally used to calibrate the INTB model. The
boxplots represent the range of mean monthly streamflow projections over eight GCMs for the
business as usual water use scenario. Retrospective GCMs (blue box plots) reproduced mean
streamflow simulated using NLDAS-2 data quite closely for both river gages with relatively
small variation among GCMs. In the dry season (October-May) future 1 (red box plots) and
future 2 (green box plots) business as usual mean monthly streamflow values over the 8 GCMs
(red box plots) also showed relatively small differences with the retrospective predictions, but
larger variation across GCMs. However in the wet season (June through September) future mean
monthly streamflows for the business as usual scenario were lower than retrospective, especially
in future 2 (2070-2100), and showed much larger variability across GCMs.

3.3 Projections of Groundwater Level

Figure 3 shows observed, NLDAS-2 predicted, and retrospective mean monthly

groundwater level for the NWH-RMP-08s (Fig. 3a), CBR-SERW-s (Fig. 3b), and STK-
STARKEY-20s wells (Fig. 3c), as well as future mean monthly groundwater level in future 1
(2030-2060) and future 2 (2070-2100) for the business as usual water use scenario and the
Hargreaves $ET_0$ method. Groundwater levels projected by retrospective GCMs showed relatively
small variation across GCMs and were very similar to groundwater levels simulated using the
historic NLDAS-2 data for all three wells. Although observed seasonal patterns were reproduced
accurately for all wells during the retrospective period, NWH-RMP-08s retrospective
groundwater level predictions were lower than observed groundwater levels throughout the year
(Fig. 3a). In contrast, all CBR-SERW-s and STK-STARKEY-20s retrospective groundwater
lever predictions were higher than observed groundwater levels throughout the year (Figs. 3b and
3c). These deviations (which are generally less than 0.5m) are consistent with deviations
between the observed data and groundwater levels simulated by the original calibrated model
using the locally-observed point weather data (Guerink and Basso, 2013). The mean groundwater
levels averaged over GCMs for the future period 1 (2030-2060) business as usual scenario were
similar to, or slightly lower than, the mean retrospective groundwater levels; however the mean
groundwater levels for future 2 (2070-2100) were significantly lower than mean groundwater
levels in the retrospective period, especially in the wet season for all wells. Similar to the
streamflow results variability in projected groundwater levels among GCMs was larger in future
2 (2070-2100) than in future 1 (2030-2060).

3.4 Changes in Future Surface Water Availability for Public Supply

Tampa Bay Water operates surface-water pumps on the Hillsborough and Alafia rivers to

help meet public water demand. The volume of flow permitted for extraction varies daily based
on maintaining sufficient in-stream flows and spring flows to protect aquatic ecosystems. In this
study, the amount of water that could be withdrawn for public water supply, while meeting
current environmental regulations, was analyzed to evaluate projected changes in future water
availability for different GCMs and  water use scenarios. Boxplots in Fig. 4a show the variation
in the projected change in the mean available water that can be withdrawn from the Hillsborough
river (the mean available water that can be withdrawn for future streamflow – the mean available
water that can be withdrawn for retrospective streamflow) over all GCMs and all $ET_0$ methods
for each water use scenario. The boxplots show large variations due to large differences in future
streamflow projections. All boxplots encompass both positive and negative changes for both
future periods, but indicate generally lower water availability in future 2 (2070-2100) than future
1 (2030-2060). Figure 4b compares the change in the projected mean available water that can be
withdrawn from the Hillsborough river over water use scenarios and $ET_0$ methods for each GCM.
While there is some variation across water use scenarios and $ET_0$ methods, Fig. 4b clearly shows
that projected changes in future surface water availability depend strongly on choice of GCM,
with 5 GCMs showing less surface water availability in the future regardless of  water use
scenario. Plots for the Alafia River show very similar behavior both by water use scenario and by
GCM (Figure S1 in supplemental materials).
The differences between the mean projected changes in available water that can be
withdrawn from the Hillsborough and Alafia rivers for individual water use scenarios over
GCMs and $ET_0$ methods (left columns in Table 5), and for individual GCMs over water use
scenarios and $ET_0$ methods (right columns in Table 5), were evaluated for statistical significance
using Tukey's HSD (honest significant difference) test. The HSD test confirmed that none of the
differences in the mean projected change in available water for different water use scenarios
shown in Figure 3a were statistically significant for the Hillsborough river for either future
period (In Table 5 scenarios with  the same alphabetic subscripts are not statistically significantly
different). For the Alafia river the mean projected changes in available water for the extreme
groundwater pumping reduction scenario was statistically significantly different from the other
water use scenarios in future 1 (2030 – 2060), but no statistically significant changes were
detected in future 2 (2070 – 2100). These results imply that due to the large variations in climate
projections produced by different GCMs, differences in mean projected changes in streamflow
projections due to differences water use scenarios and $ET_0$ methods cannot be reliably predicted
by averaging over GCMs.
On the other hand, many of the differences between mean projected changes in available
water that can be withdrawn from the Hillsborough and Alafia rivers for individual GCMs over
water use scenarios were statistically significant for both future periods (i.e. many of the GCMs
on the right side of Table 5 have different alphabetic subscripts). Two GCMs show a distinct
increase water availability from these rivers for public supply (GFDL-CM3 and MRI-CGCM3)
however, most GCMs show a decrease in water availability (BNU-ESM, GFDL-ESM2G,
MIROC-ESM, NorESM1-M, and BCC-CSM). These results underscore the fact that differences
in projections of future availability of water from these rivers for public supply are driven more
strongly by differences climate models than differences in future human water use scenarios or
$ET_0$ methods. Furthermore manipulating groundwater use to change the amount of available
surface water has a very small effect for a given climate. These results are similar to previous
studies (Bosshard et al., 2013; Forzieri et al., 2014; Guimberteau et al., 2013; Harding et al.,
2012; Kay and Davies, 2008) that showed climate models are a large source of uncertainty for
climate-impact projections because of the divergence of GCM projections.

In addition, to the HSD test, the two sample t-test was conducted to evaluate statistical

significance of differences between the mean available water that can be withdrawn for the
retrospective period and the mean available water that can be withdrawn for each future water
use scenario calculated over all GCMs and $ET_0$ methods. The two sample t-test indicated that, at
the 0.05 significance level, none of the future scenarios were statistically significantly different
from the retrospective business as usual scenario for the Hillsborough river. For the Alafia river
only the no pumping and no urban pumping scenarios in future 1 (2030-2060) showed significant
differences from the retrospective scenario in the available water that can be withdrawn from the
Alafia river (marked as † on the left hand columns of Table 5). In contrast most GCMs projected
significantly different mean available water in both future periods compared to the retrospective
period when averaged over water use scenarios (marked as † in right hand columns of Table 5).

The results that future streamflow projections are relatively insensitive to water use

scenarios are contrary to that of Dale et al. (2015). They used historical streamflow and climate
data to evaluate the impacts of anthropogenic change on streamflow and found that for an
irrigation intensive watershed located in an area with hot summer and limited precipitation
(North Central Oklahoma, U.S.) irrigation from groundwater pumping increased antecedent soil
moisture and played an equally important role in streamflow variability as climate change. These
differences are likely due to that fact that the region studied here is wetter than their study region,
the aquifer underlying the study region is large and productive, and land use changes were not
considered in this study.

3.5 Changes in Compliance with Groundwater Level Regulations

Groundwater pumping for water supply in the Tampa Bay region is regulated to maintain
groundwater levels that promote environmental protection by preventing dewatering of lakes and
wetlands near wellfields. The relative importance of water use scenario and GCM selection on
the change in percent of time that future groundwater levels were above the target levels (the
percent of the time that groundwater level is above the target level for future scenario – the
percent of the time that groundwater level is above the target level for retrospective scenario)
was evaluated for three monitoring wells. Boxplots in Fig. 5a show the change in percent of the
time that groundwater level was above the target level in the dry season (Oct – May) for the
NWH-RMP-08s well over all GCMs for each water use scenario and $ET_0$ methods. Tukey's
HSD test showed that the two most extreme water use reduction scenarios, i.e. the no pumping
scenario and the no urban pumping scenario, showed a statistically significant higher percent of
time that groundwater is above the target level in future 1 (2030-2060) compared to the other
future water use scenarios for the NWH-RMP-08s well (Table 6). Furthermore the T-test showed
a statistically significant difference in the percent of time this well was above the target level in
both futures 1 (2030-2060) and 2 (2070-2100) for these two scenarios compared to the
retrospective scenario (marked with † in Table 6). Results for the other two wells were more
ambiguous with Tukey's HSD test showing differences among several of the water use scenarios
in future 1 for both wells, and among several water use scenarios in future 2 for STK-
STARKEY-20s. The T-test for CBR-SERW-s and STK-STARKEY-20s showed statistically
significant differences for the two most extreme water use reduction scenarios compared to the
retrospective scenario both future 1 and future 2. Collectively these results confirm that future
compliance with groundwater levels is sensitive to water use scenario. Scenarios that assume
differences in CWF pumping predict statistically significant differences in future groundwater
compliance when averaged over possible future climates and $ET_0$ methods. On the other hand
scenarios that assume similar differences in the magnitude of agricultural pumping generally do
not show statistically significant differences in future groundwater compliance. These results are
largely explained by the concentration of CWF wells near monitoring wells versus the
distribution of agricultural pumping wells throughout the model domain.
Fig 5b indicates and Tukey's HSD test (Table 7) confirms that the mean change in
percent of time that groundwater is above the target level in the monitoring wells was
significantly different for many GCMs in both future periods for all three wells (Figure 5 and
Figures S2 – S3 in the supplemental material. Two "wet" GCMs (GFDL-CM3 and MRI-
CGCM3) projected statistically significant increases in the mean percent of the time that
groundwater is above the target level for both future periods compared to the retrospective period
in all three wells  when averaged over future water use scenario and $ET_0$ method(Fig. 5b and
marked as † in the Table 7). Three "drier" GCMs (BNU-ESM, MIROC-ESM and BCC-CSM)
projected statistically significant decreases in percent of the time that groundwater level is above
the target level compared to the retrospective period in future 2 (2070-2100) for all three wells.
More GCMs showed significant differences in future period 2 (2070-2100) than in future period
1 (2030-2060) compared to the retrospective period because the differences among climate
model projections increase in the later future. These results indicate that for drier future climate
groundwater level regulations may be difficult to achieve regardless of groundwater pumping
scenario, and thus may have to change with the changing climate.

3.6 Ability to Meet Future Water demand

Future water demand projections for Tampa Bay Water indicate that even with active

urban water conservation programs public water supply demand is expected to increase from
approximately 220 MGD in 2010 to approximately 278 MGD in 2045 (Tampa Bay Water Water
Demand Management Plan Final Report, 2013). At the present time the Tampa Bay water supply
system includes 90 MGD groundwater pumping permitted for the CWF, a 25 MGD desalination
plant and permitted water withdrawals from the Hillsborough and Alafia rivers that vary daily to
maintain ecologically protective in-stream flows. Scenario discovery analysis (Tariq et al., 2017)
was used to explore the ability of Tampa Bay Water to meet 2045 water demand with while
maintaining or improving existing levels of compliance with surface and groundwater
regulations.

Figure 6 presents the results of the scenario discovery analyses that evaluates which

climate and water use scenarios achieve these objectives in future 1 (2030-2060) using the
Hargreaves $ET_0$ method. In these analyses it was assumed that Tampa Bay Water's desalination
capacity would remain at 25 MGD, surface water would be extracted at the maximum rate that
complied with existing regulations, and 0% (current condition), 20%, or 40% of Tampa Bay
Water's public water supply (surface water, groundwater, and desalination) might be reclaimed
and reused to satisfy public demand. The axes in figure 6 represent the two most important
factors in the climate and water use scenarios that affect achievement of Tampa Bay Water's
goals:  mean change in precipitation projected by the different GCMs and volume of agricultural
and urban groundwater pumping in the water use scenario. Green filled circles indicate futures
that meet both 2045 water demand and maintain groundwater compliance levels at or above
current conditions in future 1 (2030-2060). Yellow filled circles indicate futures that meet 2045
water demand but decrease the level of groundwater compliance. Orange filled circles indicate
futures that do not meet 2045 water demand but maintain groundwater compliance levels at or
above current conditions. Red filled circles indicate futures that do not meet 2045 water demand
and decrease the level of groundwater compliance. The black filled circle indicates the
retrospective business as usual condition.

Figure 6a shows that, without using reclaimed water to satisfy public water demand only

4 scenarios are able to meet 2045 demand and maintain or improve existing levels of compliance
with groundwater regulations (filled green circles on Fig 6a).  These 4 scenarios assume the 2
wettest future climates (projected by GFDL-CM3 and MRI-CGM3) will occur and permitted
CWF pumping will increase from 90 MGD to 130 MGD.  No other climate-water use scenarios
are able to meet 2045 demand without use of reclaimed water (there are no yellow filled circles
on Fig. 6a).  In fact a significant number of the scenarios, including many that assume the
business as usual water use scenario, are not able to meet 2045 demand and also decrease
compliance groundwater regulations (red filled circles on Fig 6a).

Figure 6b shows that 20% of freshwater withdrawn is reclaimed and used to satisfy

public demand the two wettest future climates can meet 2045 demand and maintain or improve
existing levels of compliance with groundwater regulations for all water use scenarios. However
no other scenarios are able to achieve both goals. If 40% of freshwater withdrawn is reclaimed
and used to satisfy public demand more scenarios are able to achieve both goals. These scenarios
include the climate scenarios that project that at least the existing average annual rainfall will
occur in the future (i.e. projected change in average annual rainfall greater than or equal to zero).
However to meet both public water demand and maintain existing compliance with groundwater
regulations, scenarios that predict the same rainfall as current climate require a complete switch
of public water supply from groundwater to surface water sources (bottom two water use
scenarios in Fig 6). This would require Tampa Bay Water to significantly increase their surface
water storage and treatment capacity and eliminates the use of their most inexpensive water
source (groundwater). If groundwater regulations were relaxed, and 40% freshwater withdrawn
in reclaimed, 2045 demand could be met under any climate scenario (yellow circles in Fig. 6c). It
should be noted that the Regional Water Supply Planning (2016) reported that in 2015 only
about 11.5% of total freshwater withdrawn was reused in Florida. Therefore reclaiming 20% -
40% of freshwater withdrawn will be a significant investment.

**4 Conclusions**

It is important to evaluate possible changes in future streamflow and groundwater levels

to evaluate risks in water resources management and planning. This study investigated potential
future changes in hydrologic systems, ability to meet future water demand, and compliance with
water resource regulation using eight GCMs, eight human water use scenarios and three $ET_0$
methods to drive an integrated hydrologic model developed for the Tampa Bay region.
Variance-based sensitivity analysis showed that changes in projected streamflow were very
sensitive to GCM selection, but relatively insensitive $ET_0$ method or water use scenario. Changes
in projections of groundwater level were sensitive to both GCM and water use scenario, but
relatively insensitive to $ET_0$ method.

The eight GCMs projected diverse changes in streamflow and groundwater level, with

most GCMs projecting statistically significant different future streamflow and groundwater
levels than the current condition. Five of the 8 GCMs projected a decrease in future streamflow
and groundwater level in the INTB region regardless of water use scenario or ET method.  None
of the 8 GCMs projected that 2045 water demand could be met under the business as usual water
use scenario. Two GCMs (GFDL-CM3 and MRI-CGCM3) predicted increased streamflow and
groundwater levels and an ability to meet 2045 projected water demand and maintain existing
levels of compliance with groundwater standards if permitted CWF pumping were increased
from the current 90 MGD to 130 MGD. The GCM that predicted that future annual average
rainfall will be approximately equal to current rainfall met 2045 demand maintained existing
levels of compliance with groundwater standards only for the water use scenarios that eliminated
CWF pumping completely and reclaimed 40% of freshwater withdrawals.

These results suggest that it is more likely than not that climate change will reduce the

availability of both surface and groundwater for public supply in the Tampa Bay Region. Current
regulations on water withdrawals (surface water withdrawal permit thresholds and target levels
in monitoring wells near lakes and wetlands) may have to adapt to future climate conditions
since only extreme changes human water use (i.e. dramatic increases in use of reclaimed water
and a complete switch from groundwater to surface water) may be able to maintain retrospective
hydrologic regimes and associated aquatic ecosystems and meet human water demand in the
future.

It should be noted that the findings of this study are limited by a few major assumptions.

For example this study used only 8 GCMs to project future climate which is a relatively small
number. However these 8 GCMs spanned the range of cool to warm bias and wet to dry bias
exhibited by 41 CMIP5 GCMs for the southeastern United States (Rupp, 2016). In addition land
use change was not considered in this study. Instead we assumed the increases in agricultural and
urban water demand were the result of intensification of water use on existing land uses. This
assumption is consistent with a regional planning strategy that promotes agricultural and urban
intensification on existing lands, along with protection of existing conservation lands, wetlands
and water supplies (Barnett et al., 2007). However future work should build on this study to
evaluate the additional impacts of potential land use change scenarios (Gupta et al., 2015; Lin et
al., 2015; Matheussen et al., 2000; Yan et al., 2013).

**Acknowledgments**

This research was supported by Tampa Bay Water and the University of Florida Water

Institute. We gratefully acknowledge the modeling groups participating in the Program for
Climate Model Diagnosis and Inter-comparison (PCMDI) for their role in making the CMIP5
(Coupled Model Intercomparison Project) multi-model data set available.

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

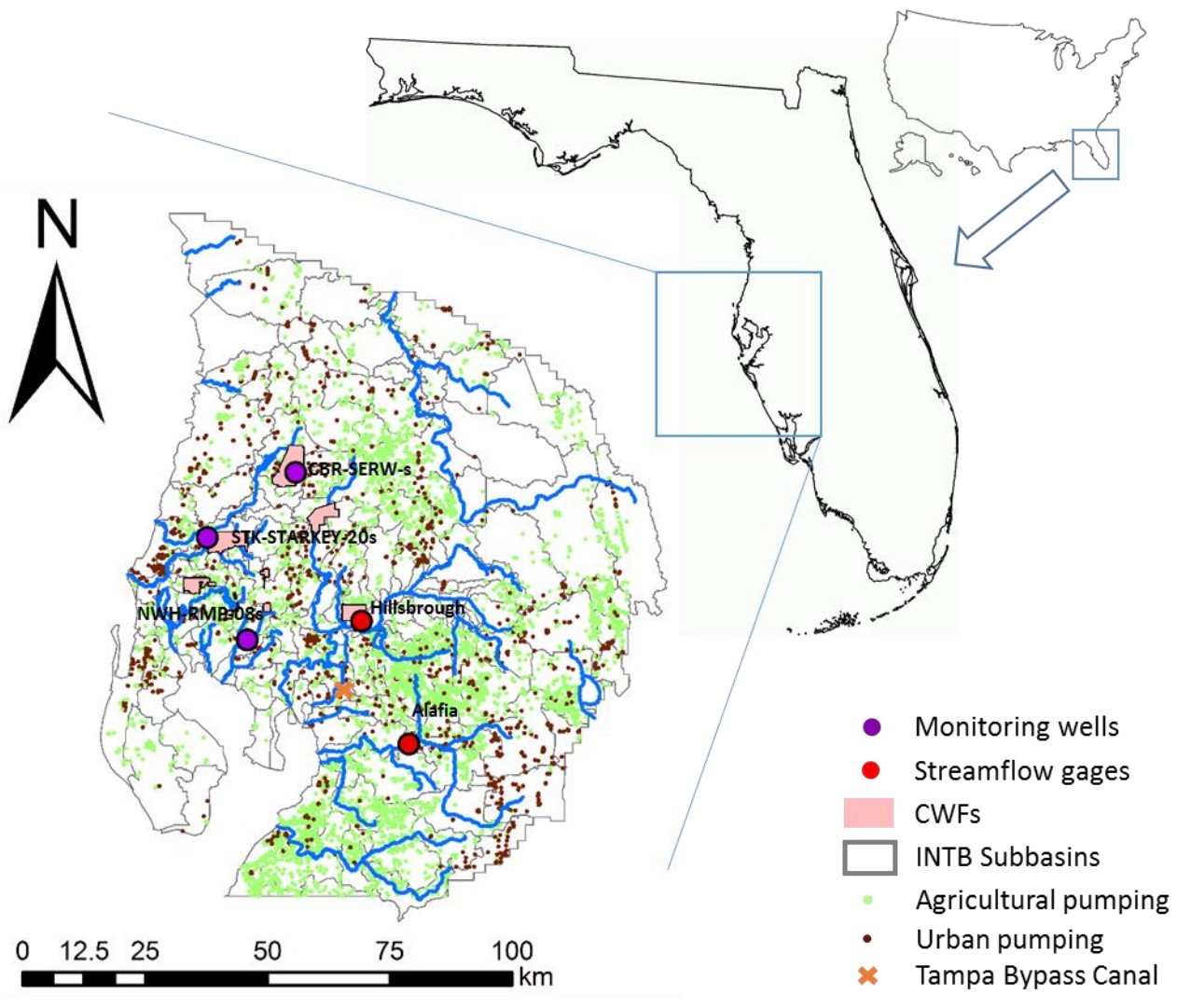


Figure 1. Study region showing the INTB model domain and locations of agricultural, industrial

and public water supply wells, the Tampa Bay Waters Consolidated Wellfields (CWF), two

streamflow locations where water is withdrawn for public supply, the Tampa Bay Bypass Canal,

and three  monitoring wells near Tampa Bay Water's CWFs that are used to evaluate compliance

with groundwater level regulations.


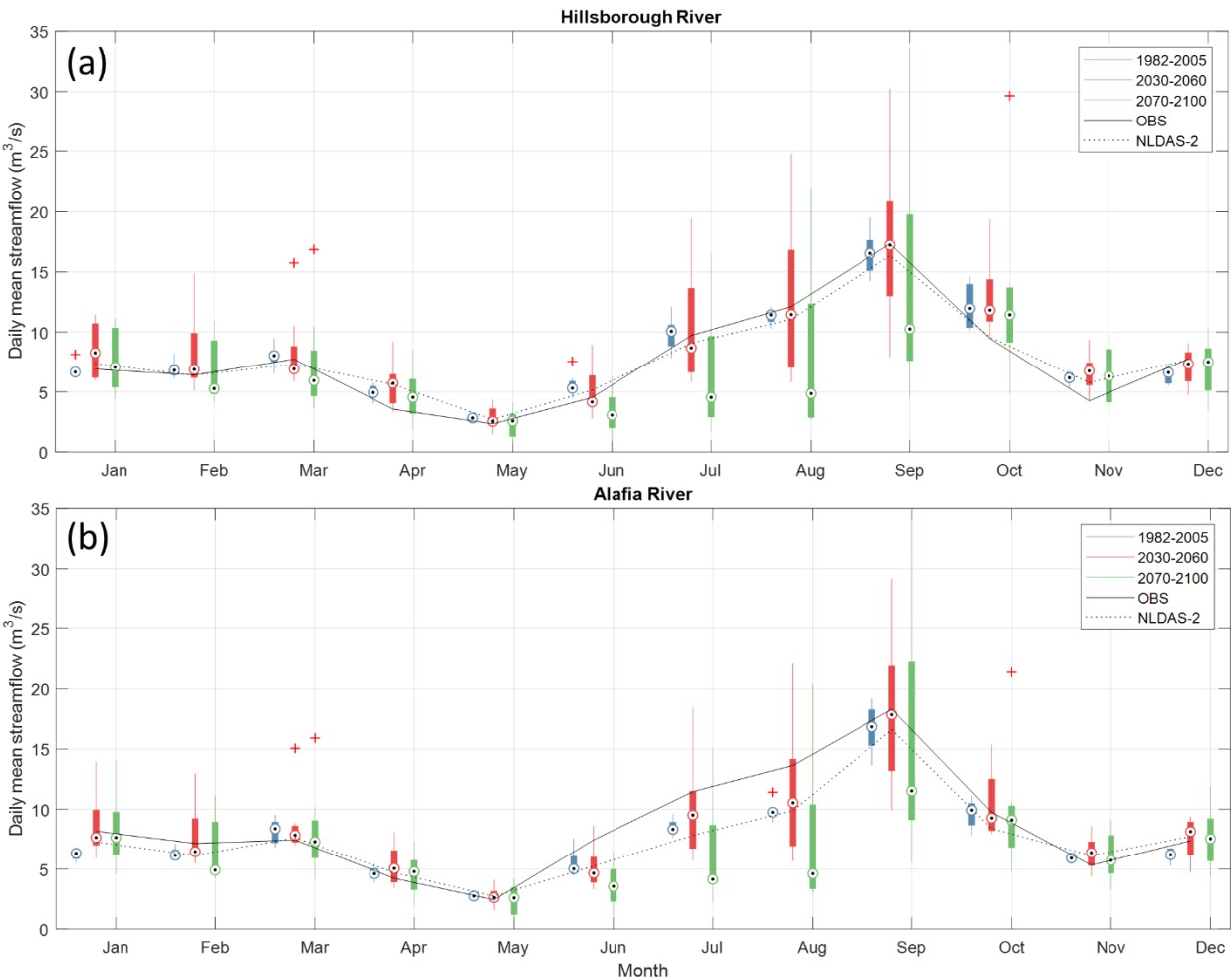


Figure 2. Mean monthly streamflow for the Hillsborough river (top) and Alafia river (bottom) for
business as usual scenario water use and Hargreaves $ET_0$ method. Box plots indicate range of
prediction over the 8 GCMs.

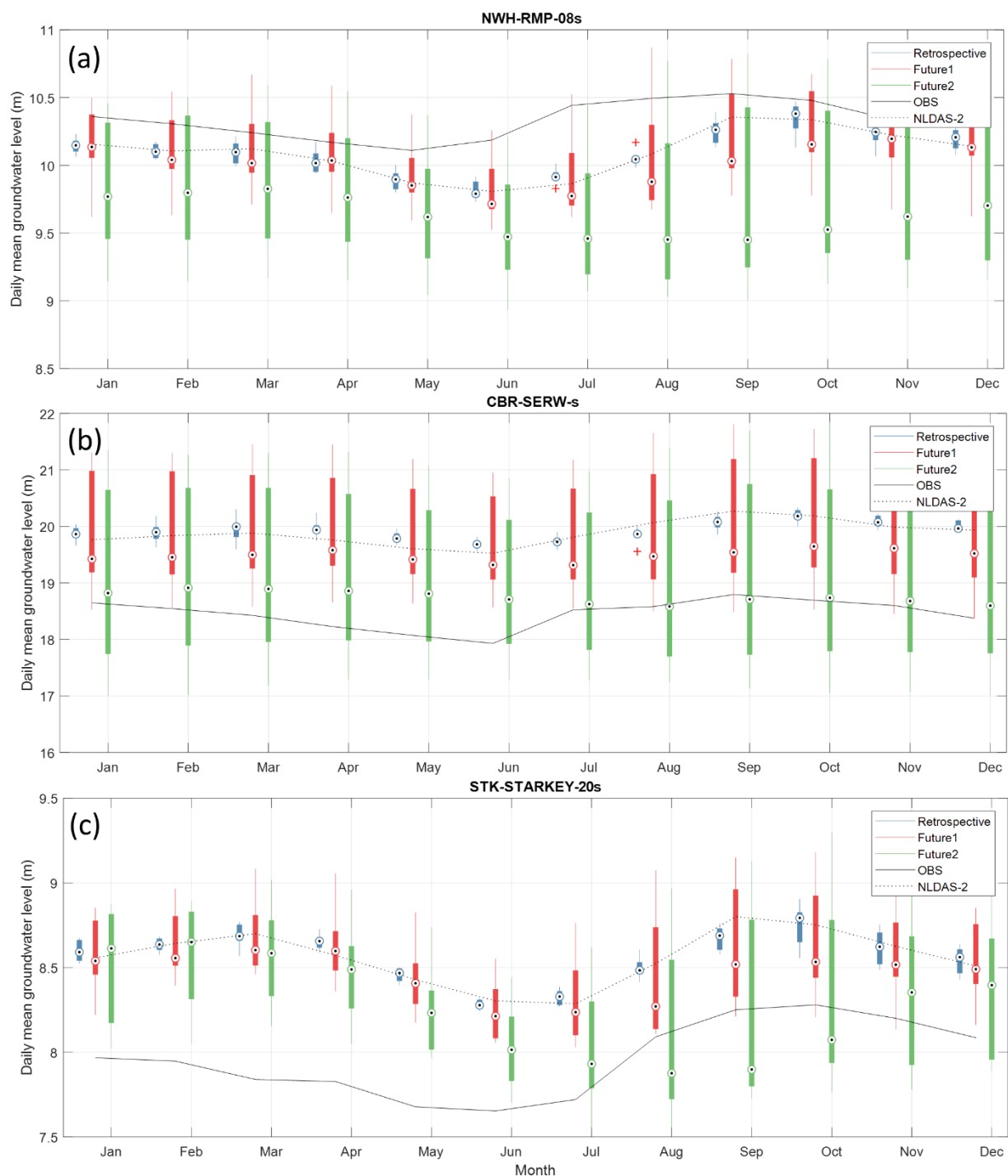


Figure 3. Mean monthly groundwater level for the NWH-RMP-08s (top), CBR-SERW-s
(middle) and STK-STARKEY-20s (bottom) for business as usual water use scenario and
Hargreaves $ET_0$ method. Box plots indicate range of prediction over the 8 GCMs.


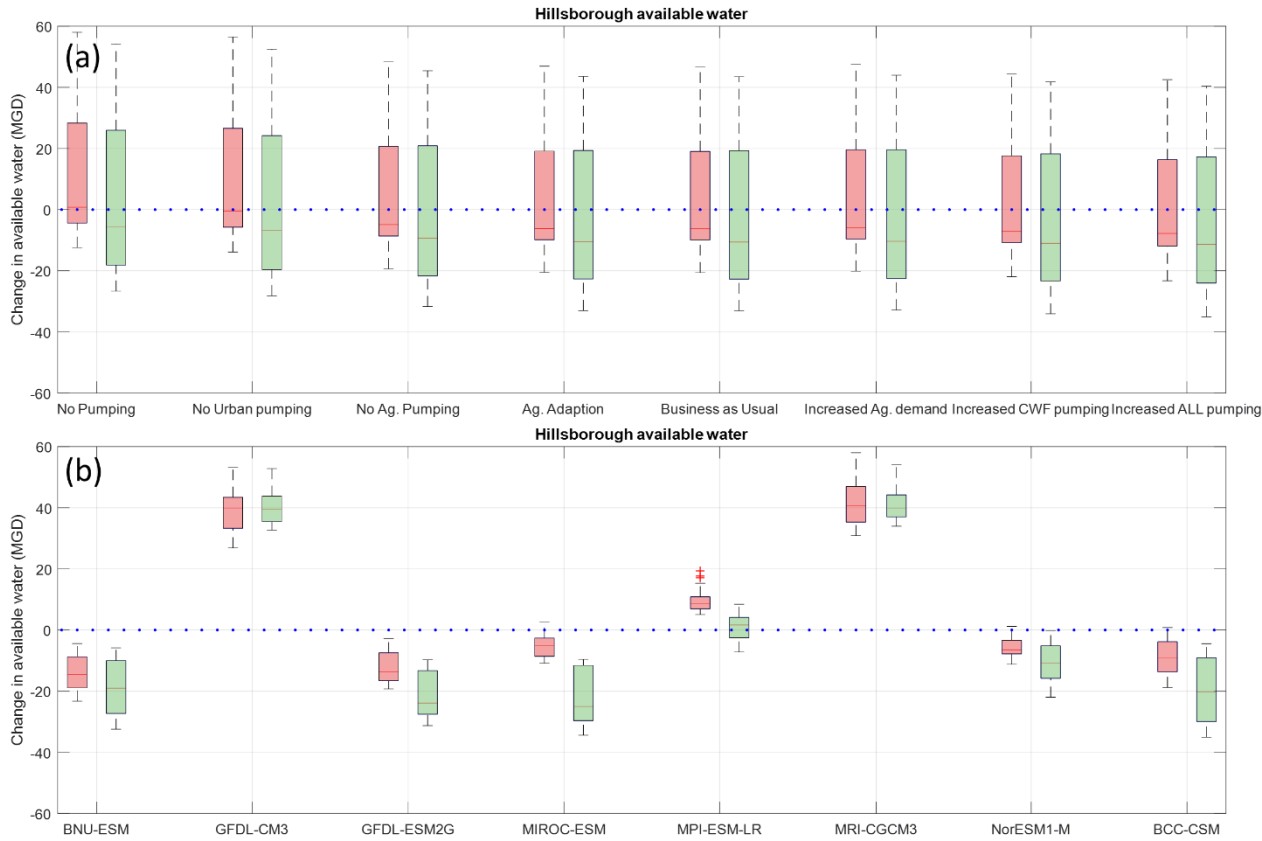


Figure 4. The change in amount of available water can be withdrawn from Hillsborough river by
(a) different water use scenarios over GCMs and $ET_0$ methods and by (b) different GCMs over
water use scenarios and $ET_0$ methods.

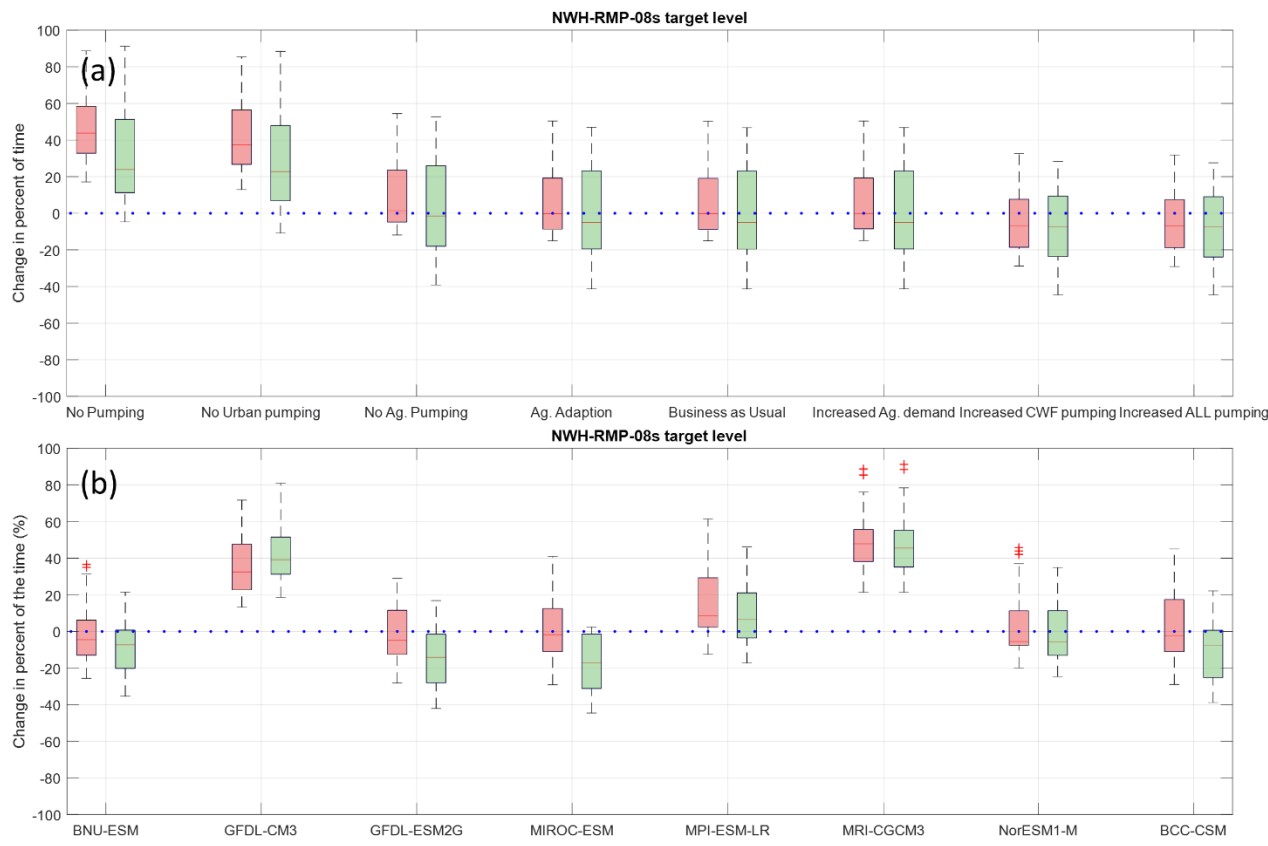


Figure 5. The change in the percent of the time that groundwater level is above the target level
for NWH-RMP-08s well by (a) different water use scenarios over GCMs and $ET_0$ methods and
by (b) different GCMs over water use scenarios and $ET_0$ methods.

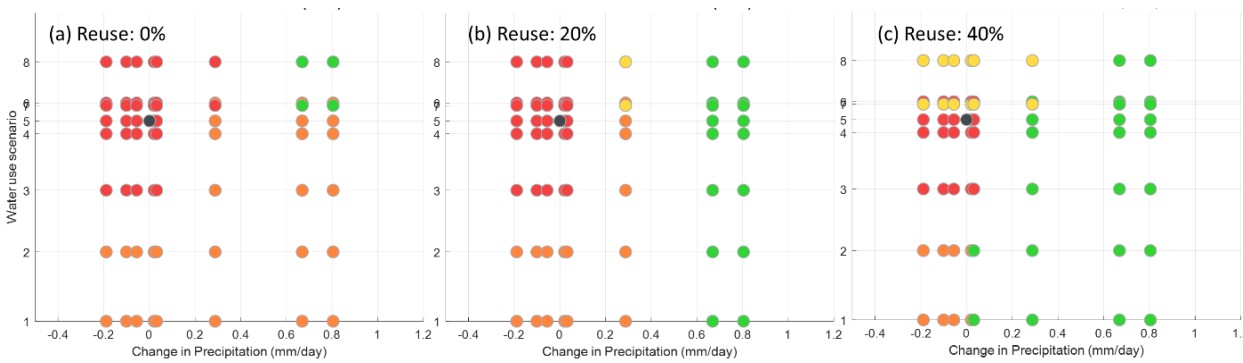


Figure 6. Scatterplot of futures in which the Tampa Bay Water meets 2045 water demands and maintains or improves compliance with groundwater regulations in future 1 (2030-2060) assuming 0%, 20% and 40% of freshwater withdrawn is reclaimed and reused to satisfy urban demand. Green filled circles indicate futures that meet both 2045 water demand and maintain groundwater compliance levels at or above current conditions. Yellow filled circles indicate futures that meet 2045 water demand but decrease the level of groundwater compliance. Orange filled circles indicate futures that do not meet 2045 water demand but maintain groundwater compliance levels at or above current conditions. Red filled circles indicate futures that do not meet 2045 water demand and decrease the level of groundwater compliance. The black filled circle indicates the retrospective business as usual condition.


**Table 1.** Description of the CMIP5 models used in this study.

| Model | Institute (country) | Resolutions | Calendar | ΔPrecipitation (mm/yr)* | | ΔET$_0$ (mm/yr)* | | Reference |
|---|---|---|---|---|---|---|---|---|
| | | | | 2030-2060 | 2070-2100 | 2030-2060 | 2070-2100 | |
| (1) BNU-ESM | College of Global Change and Earth System Science, Beijing Normal University (China) | 2.8° lat × 2.8° lon | No leap | -68.9 | -57.1 | 93.3 | 273.5 | Ji et al. (2014) |
| (2) GFDL-CM3 | NOAA/Geophysical Fluid Dynamics Laboratory (USA) | 2.0° lat × 2.5° lon | No leap | 293.6 | 400.0 | 133.1 | 351.5 | Guo et al. (2014) |
| (3) GFDL-ESM2G | NOAA/Geophysical Fluid Dynamics Laboratory (USA) | 2.0° lat × 2.5° lon | No leap | -36.8 | -134.6 | 56.2 | 133.5 | Taylor et al. (2012) |
| (4) MIROC-ESM | Atmosphere and Ocean Research Institute, National Institute for Environmental Studies, and Japan Agency for Marine-Earth Science and Technology (Japan) | 2.8° lat × 2.8° lon | Leap year | 7.5 | -153.9 | 99.9 | 240.8 | Watanabe et al. (2011) |
| (5) MPI-ESM-LR | Max Planck Institute for Meteorology (Germany) | 1.87° lat × 1.87° lon | Leap year | 105.1 | 77.8 | 81.8 | 230.9 | Block and Mauritsen (2013) |
| (6) MRI-CGCM3 | Meteorological Research Institute (Japan) | 1.12° lat × 1.12° lon | Leap year | 244.2 | 281.2 | 24.4 | 122.1 | Yukimoto et al. (2012) |
| (7) NorESM1-M | Norwegian Climate Centre (Norway) | 1.9° lat × 2.5° lon | No leap | 11.6 | 3.0 | 137.7 | 324.6 | Bentsen et al. (2013) |
| (8) BCC-CSM1.1 | Beijing Climate Center (China) | 2.8° lat × 2.8° lon | No leap | -20.4 | -117.5 | 118.1 | 303.6 | Xiao-Ge et al. (2013) |

* Change in precipitation (or ET$_0$) is defined as average of future period minus average of retrospective period.


**Table 2.** Future scenario summary

| Scenario Name | Scenario Number | Irrigation Applied to Land Surface | Agricultural pumping | Urban pumping |
|---|---|---|---|---|
| No pumping | 1 | No | No | No |
| No urban pumping | 2 | AFSIRS[*] | 85 % efficiency | No |
| No agricultural pumping | 3 | No | No | RETRO [**] CWF 13 mm/yr, Total 51 mm/yr |
| Agricultural adaption | 4 | AFSIRS | 85 % efficiency Groundwater pumping offset by 6 mm/yr | RETRO CWF 13 mm/yr, Total 51 mm/yr |
| Business as Usual | 5 | AFSIRS | 85 % efficiency | RETRO CWF 13 mm/yr, Total 51 mm/yr |
| Increased agricultural demand | 6 | Increased by 6 mm/yr | 85 % efficiency | RETRO CWF 13 mm/yr, Total 51 mm/yr |
| Relaxed regulatory requirements for urban pumping | 7 | AFSIRS | 85 % efficiency | Increase CWF by 6 mm/yr to 19 mm/yr CWF 19 mm/yr, Total 57 mm/yr |
| Relaxed regulatory requirements for all pumping | 8 | AFSIRS | 85 % efficiency | Increase all urban pumping by 130/90 CWF 19 mm/yr, Total 74 mm/yr |

\* AFSIRS: climate driven irrigation water demand estimated by AFSIRS model using GCMs.
\*\* RETRO: groundwater pumping in the future will be equal to retrospective groundwater pumping.



**Table 3.** The first order sensitivity index of change in streamflow (future – retrospective period).

| River gage | Season | Period | GCM | $ET_0$ | Water use scenario |
|---|---|---|---|---|---|
| Hillsborough | Wet season | 2030-2060 | 0.944 | 0.002 | 0.016 |
| | | 2070-2100 | 0.940 | 0.041 | 0.006 |
| | Dry season | 2030-2060 | 0.948 | 0.012 | 0.029 |
| | | 2070-2100 | 0.961 | 0.001 | 0.018 |
| Alafia | Wet season | 2030-2060 | 0.928 | 0.010 | 0.031 |
| | | 2070-2100 | 0.952 | 0.021 | 0.012 |
| | Dry season | 2030-2060 | 0.876 | 0.012 | 0.072 |
| | | 2070-2100 | 0.927 | 0.001 | 0.068 |



**Table 4.** The first order sensitivity index of change in groundwater level (future – retrospective
period).

| Monitoring well | Season | Period | GCM | ET$_0$ | Water use scenario |
|---|---|---|---|---|---|
| NWH-RMP-08s | Wet season | 2030-2060 | 0.442 | 0.005 | 0.501 |
| | | 2070-2100 | 0.576 | 0.004 | 0.278 |
| | Dry season | 2030-2060 | 0.475 | 0.007 | 0.435 |
| | | 2070-2100 | 0.550 | 0.002 | 0.288 |
| CBR-SERW-s | Wet season | 2030-2060 | 0.656 | 0.000 | 0.214 |
| | | 2070-2100 | 0.755 | 0.002 | 0.143 |
| | Dry season | 2030-2060 | 0.639 | 0.001 | 0.221 |
| | | 2070-2100 | 0.747 | 0.002 | 0.146 |
| STK-STARKEY-20s | Wet season | 2030-2060 | 0.604 | 0.000 | 0.325 |
| | | 2070-2100 | 0.718 | 0.004 | 0.198 |
| | Dry season | 2030-2060 | 0.584 | 0.002 | 0.330 |
| | | 2070-2100 | 0.707 | 0.001 | 0.200 |



**Table 5.** The results of Tukey's HSD test of mean change in amount of available water (MGD)
that can be withdrawn from Hillsborough river or Alafia river for each water use scenario over
GCM and $ET_0$ method, or for each GCM over water use scenario and $ET_0$ method (Comparison
of all possible pairs of means).

| By human water use scenario | Hillsborough | | Alafia | | By GCM | Hillsborough | | Alafia | |
|---|---|---|---|---|---|---|---|---|---|
| | 2030-2060 mean | 2070-2100 mean | 2030-2060 mean | 2070-2100 mean | | 2030-2060 mean | 2070-2100 mean | 2030-2060 mean | 2070-2100 mean |
| No Pumping | 11.63 a | 3.88 a | 4.89 a[†] | 2.28 a | BNU-ESM | -14.03 e[†] | -18.76 d[†] | -4.25 d[†] | -5.89 c[†] |
| No Urban Pumping | 10.10 a | 2.61 a | 4.00 a[†] | 1.45 a | GFDL-CM3 | 39.20 a[†] | 40.27 a[†] | 8.16 a[†] | 9.11 a[†] |
| No Ag. Pumping | 5.57 a | -1.21 a | 1.48 a | -0.99 a | GFDL-ESM2G | -12.24 de[†] | -21.68 d[†] | -1.84 cd | -5.70 c[†] |
| Ag. Adaption | 4.22 a | -2.54 a | 0.85 ab | -1.60 a | MIROC-ESM2G | -5.01 c | -22.31 d[†] | -0.09 c | -6.26 c[†] |
| Business as Usual | 4.16 a | -2.59 a | 0.82 ab | -1.63 a | MPI-ESM-LR | 9.71 b[†] | 1.07 b | 2.01 b | -0.56 b |
| Increased Ag. Demand | 4.56 a | -2.27 a | 1.00 ab | -1.47 a | MRI-CGCM3 | 41.64 a[†] | 41.34 a[†] | 10.64 a[†] | 10.46 a[†] |
| Increased CWF pumping | 2.90 a | -3.66 a | 0.81 ab | -1.64 a | NorESM1-M | -5.58 c | -10.71 c[†] | 0.78 bc | -2.21 c[†] |
| Increased All Pumping | 1.72 a | -4.65 a | -0.43 b | -2.73 a | BCC-CSM | -8.84 cd[†] | -19.67 d[†] | -1.98 cd | -5.28 c[†] |

Means with different subscripts were significantly different in Tukey's HSD test.

[†]: The results were significantly different than retrospective BAU by two sample t-test at the 0.05 significance level.



**Table 6.** The results of Tukey's HSD test of mean change in the percent of the time that
groundwater level is above the target level for monitoring wells over all GCMs and $ET_0$ methods
for each water use scenario (Comparison of all possible pairs of means).

| By human water use scenario | NWH-RMP-08s | | CBR-SERW-s | | STK-STARKEY-20s | |
|---|---|---|---|---|---|---|
| | 2030-2060 mean | 2070-2100 mean | 2030-2060 mean | 2070-2100 mean | 2030-2060 mean | 2070-2100 mean |
| No Pumping | 46.04 a[†] | 32.21 b[†] | 31.93 a[†] | 22.79 a[†] | 27.87 a[†] | 18.00 a[†] |
| No Urban Pumping | 41.17 a[†] | 28.36 a[†] | 31.40 ab[†] | 22.45 a[†] | 26.91 ab[†] | 17.22 ab[†] |
| No Ag. Pumping | 10.28 b | 3.69 b | 11.00 c[†] | 7.21 a | 3.92 a[†] | -2.04 bc |
| Ag. Adaption | 6.66 b | 0.88 b | 10.76 c | 7.06 a | 3.15 ab | -2.79 c |
| Business as usual | 6.55 b | 0.81 b | 10.73 c | 7.04 a | 3.12 ab | -2.80 c |
| Increased Ag. Demand | 6.70 b | 0.89 b | 11.14 bc[†] | 7.32 a | 3.21 ab | -2.73 c |
| Increased CWF pumping | -4.25 b | -7.81 b | 5.23 c | 3.01 a | -4.31 b | -9.05 c |
| Increased All Pumping | -4.64 b | -8.13 b | 4.08 c | 1.93 a | -6.07 b | -10.52 c[†] |

Means with different subscripts were significantly different in Tukey's HSD test.
[†]: The results were significantly different than retrospective BAU by two sample t-test at the 0.05 significance level.

**Table 7.** The results of Tukey's HSD test of mean change in percent of the time that
groundwater level is above the target level for monitoring wells over all water use scenarios and
$ET_0$ methods for each GCM (Comparison of all possible pairs of means).

| By GCM | NWH-RMP-08s | | CBR-SERW-s | | STK-STARKEY-20s | |
|---|---|---|---|---|---|---|
| | 2030-2060 mean | 2070-2100 mean | 2030-2060 mean | 2070-2100 mean | 2030-2060 mean | 2070-2100 mean |
| BNU-ESM | -6.39 c | -18.59 bc[†] | -12.08 c[†] | -16.66 c[†] | -12.55 d | -18.30 def[†] |
| GFDL-CM3 | 32.35 ab[†] | 39.44 a[†] | 48.12 ab[†] | 56.39 a[†] | 19.56 ab[†] | 24.50 ab[†] |
| GFDL-ESM2G | -3.22 bc | -18.93 bc | -7.58 c | -22.84 c[†] | -12.96 d[†] | -16.40 cde[†] |
| MIROC-ESM | -4.83 c | -35.79 c[†] | 4.97 c | -15.52 c[†] | -12.96 d[†] | -39.01 f[†] |
| MPI-ESM-LR | 11.26 abc | 3.41 b | 29.83 b[†] | 14.15 b[†] | 12.02 abc[†] | 4.06 bc |
| MRI-CGCM3 | 41.27 a[†] | 39.67 a[†] | 62.87 a[†] | 56.38 a[†] | 34.45 a[†] | 26.16 a[†] |
| NorESM1-M | 3.84 bc | -3.47 b | 1.18 c | -8.40 c | 2.31 bcd | 0.17 cd |
| BCC-CSM | -2.38 bc | -25.30 bc[†] | 1.17 c | -12.51 c[†] | -11.45 cd | -28.99 ef[†] |

Means with different subscripts were significantly different in Tukey's HSD test.
[†]: The results were significantly different than retrospective BAU by two sample t-test at the 0.05 significance level.