# Peer review of "These supplemental figures are for section 3.2 (Figures S1 to S3) and section 3.3 (Figures S4 to"

_Hydrology and Earth System Sciences, 2018_

## Referee Comment (RC1) · Anonymous Referee #1 · 5 Apr 2018

General Comments:

Overall, I found this work to be a well-written case study that connected climate change (in the form of general circulation model data) and water use (a number of demand scenarios) to surface and ground water availability in the Tampa Bay area in Florida, USA. In its current form this work represents a small but necessary contribution to the literature; many works have forced regional hydrologic models with downscaled GCM data, but fewer have done so under different future demand scenarios and this work furthers our understanding of climatic influence on hydrologic processes of importance to humans.

I have some concerns that the chief result of this paper – climate change will have stronger influence over water availability than human use/demand and likely result in

reduced water availability regionally – was not strongly demonstrated. The authors use high sensitivity of projected water levels in the study region to downscaled GCM data as evidence of climate change's stronger influence over water availability than other factors. However, the paper references past impact analyses that claim GCM selection can have "unacceptable influence" over results, but have not provided any evidence that the eight GCMs used in this analysis do not have the same effect. Furthermore, sensitivity of water levels is based on relative variance of the contributing factors, two of the three factors (reference ET, water use scenarios) being deterministic and limited in scope and the third (downscaled GCM precipitation) randomly sampled from thousands of data points; I question whether the forms of data driving this analysis combined with the use of relative variance to determine sensitivity does not overly influence the study conclusions that GCMs have the most influence on water levels.

Beyond these larger worries, I would like to see a discussion of the results, which was largely missing from the manuscript. For two study limitations in particular, potential influence of land use and landcover change on hydrologic interactions related to water availability and the ramifications of measuring water availability without accounting for concurrent water demands, it would be good to spend more time on.

Specific Comments:

Lines 83-84: you mention in the introduction that analyses of hydrologic impacts at a catchment scale relying on a limited number of GCM projections are overly influenced by the choice of projection (Line 72) but drive your results with only 8 GCMs selected based on their provision of evapotranspiration parameters; how have you accounted for any bias in your study results from using this small subset of GCMs?

Lines 99-100: can you elaborate here? In what manner do surface waters and groundwater interact?

Lines 118-119: do you anticipate significant land use/landcover change in the Northern Tampa Bay domain relative to 1989-2006 conditions used to calibrate and validate the

[Figure]

INTB model? Do the hydrologic responses presented in this work reflect expectations of future landcover change or a maintenance of current landcover? If the answer is the latter, as explained in lines 211-213, how do you think your results would differ if future landcover change projections were included?

Lines 154-156, Table 1: aside from citations and resolution, this table has very little information about each GCM used. Perhaps you could include the statistics on lines 157-161 in this table rather than the body of the manuscript? Are there other relevant factors that differentiate each GCM, some that may help explain the wide ranges in precipitation and evapotranspiration?

Line 163: a paragraph of the introduction was devoted to the limitations of using GCMs to drive regional hydrologic models, and various downscaling techniques have been developed to address this. How does the downscaling and bias correction approach used here compare with other methods?

Lines 227-229: I am confused between this statement and the demand scenarios laid out on the following page. Do these lines mean that 2003-2009 historical averages of reservoir withdrawal rate for Tampa and TB Water use are consistent through each simulation period? Should these vary with demand scenario as demand increases or decreases? Is the daily average rate referenced here sensitive to seasonal trends, or a flat average year-round?

Line 248: what is the basis for this ratio assumption?

Lines 253-259: are scenarios 1-3 included to isolate regional factors of water availability besides pumping? Otherwise, these scenarios do not seem plausible/necessary. Some justification for their inclusion or utility in this paragraph would be helpful.

Lines 260-271, Table 2: for scenarios with changes in pumping, where do these increases/decreases come from? Are these tied to regional planning reports or other projections of regional use? It would be good to have a citation or explanation of each.

Lines 312-314, Table 4: why is there such a large difference in sensitivity to water use scenario between NWH-RMP-13s and NWH-RMP-08s? You say here that the 13s monitoring well is furthest from well fields, but it appears from Figure 1 the two NWH-RMP stations are relatively close together and both far from the nearest consolidated well field.

Lines 178-180: based on my understanding each of the eight water use scenarios projections, based on fixed historic pumping rates, and the three evapotranspiration methods, calculated for each GCM used but are not downscaled, are deterministic and limited in number, while the downscaled GCM precipitation is randomly sampled monthly from thousands of realizations with more spatial and temporal variation than either water use or ET. As a result, I would assume that the variation contributed to streamflow and groundwater levels from precipitation (GCMs) is much greater than that from evapotranspiration (ET0) or water use scenarios. Given that your method for assessing water availability sensitivity to each of these factors is based on what amounts to a normalized summation of total variance, how much variance is inherent in the GCM precipitation, ET, and water use scenarios used? Will your sensitivity calculation be biased toward identifying GCMs/precipitation as the largest contributor of variance in water availability because of the relatively large variance within the downscaled data? If this study was driven using randomly-sampled monthly water demands and fixed precipitation projections, would GCMs still be the largest driver of uncertainty?

Lines 321-334: much of this paragraph belongs in the methods section when the evapotranspiration methods are introduced. There are no actual results in this paragraph. I am curious, however, what differences are there between the ET methods this study retained and Chang et al. (2016b) used that would result in such large differences in sensitivity? Furthermore, if you only plan to elaborate on results related to a single ET method (lines 330-334), and future streamflow and groundwater levels are collectively insensitive to the three methods you selected, why is not only one ET method used

for the entire work? Perhaps the surface flow/GW sensitivity analysis related to ET methods is best left for an appendix, with a single method used as a focus in the body of the paper.

Lines 349-350, Figure 2: why are future 2 (should consider referring to this as the 2070-2100 future period; were future 1 and future 2 periods explicitly defined in the methods?) streamflows on average lower than future 1 predictions, after future 1 mean daily streamflows in Figure 2 appear to be greater than observed mean flows? It might be more informative to condense the eight sub-plots of Figure 2 in a way that better communicates the differences in monthly flow averages between water use scenarios. Could you, for instance, aggregate these plots seasonally and then have four plots, one per season, that each has eight boxplot ranges, one per scenario. It would be much easier to see the differences, I think.

Lines 360-364: how does groundwater pumping result in lower streamflows? This relates to my previous comment about lines 99-100.

Lines 403-409: I would move these lines into the methods section; I cannot find a substantial mention of this analysis angle before this section.

Lines 406-409, Figures 6 and 7: an evaluation of water availability under these criteria may not account for the seasonal patterns of demand or timely needs for water supply. Does municipal water demand fluctuate seasonally in this region, and, like other regions in the Southeastern US, do peak months of demand (summer) correspond with times with lower streamflows? Evaluating water availability for urban supply solely based on in-stream water availability and capacity constraints of surface water intakes will tell you that water is constantly available during winter and spring (high flows), but this is also when demand is down and so availability is not as crucial. Aggregating availability without accounting for concurrent demand at an annual scale as done in Figures 6 and 7 will (1) obscure the seasonal differences of availability and (2) not offer a sense of water availability when it is most needed.

Lines 452-453: to my understanding, your results show that the choice of GCM used to project water availability is the dominant cause of projection uncertainty, much more so than different water use scenarios. This is not the same as saying that climate change will drive water availability more than human use. As mentioned in the following lines (454-457), this work has shown the large uncertainty associated with GCM selection but in doing so has not shown that climate change is clearly more influential than human water demand in determining water availability in rivers, just that it is uncertain. This is especially true given that not all anthropogenic influences on water availability have been projected here (land use change, as an example), and that you have instances within your results (Table 4) where groundwater availability was more sensitive to the use scenario than to the GCM used.

Technical Corrections:

Introduction: parse out citations to show which studies emphasize which results, rather than blanket statements with many citations at the end and no citations for more specific findings or contributions

Line 78: consider adding "Furthermore, the effects of climate change..." to help the connectivity of this paragraph

Figure 1: it is difficult to read the well and gage labels. Can they be called out more effectively? Also, you mentioned this region contained multiple large municipalities – can you include municipal extents to illuminate what fraction of the region is more likely urban cover?

Line 185: add a comma and remove "and" from "Warming temperatures and reduce precipitation..." ("warming temps, reduced precip...")

Line 186: remove the comma

Lines 195 and 196: use of "lumped" is colloquial, replace with "included" or "combined"

Line 227: diversion not diversions

Tables 3 and 4: replace "Fut1" and "Fut2" with the time periods of each future simulation.

Figure 2: differences on each sub-plot are so small I had to zoom in several levels before it was noticeable. It may be more effective to convert this graph so that it fits an entire page, and remove the top portion of each graph window (y-axis values of 35-45)

Figures 6-8: increase fonts and boxplot sizes

Line 395: what streamflow projections? Is this a reference to GCM projections of streamflow in futures 1 and 2?

Lines 419-420: Abstract says 6 of 8 GCMs project less water availability, here it says 5 of 8 and Figures 6 and 8 appear to confirm that. Adjust as necessary. After reading further I understand these ratios change between groundwater and surface water availability, but this was still confusing to read to me.

Lines 445-446: The sentence "For both gages more GCMs in future period 2 were significantly different from the retrospective period than future period 1" is confusing. Consider adjusting.

---

## Referee Comment (RC2) · Anonymous Referee #2 · 16 Apr 2018

Summary: The authors present a case study analysis for the Tampa Bay region seeking to clarify the relative influence of climate change and human water use on the region's streamflows and groundwater levels. In particular, key monitored groundwater levels are a source of regulation that can constrain human abstractions for water supply. The paper is not methodologically a significant departure from the authors' prior published work. The claimed primary contribution is a variance decomposition-based global sensitivity analysis to attribute if downscaled climate change scenarios, human use scenarios, or reference evapotranspiration methods are dominant in projected trends or changes in streamflow or groundwater. Also the trends themselves and how they potentially constrain water supply are also discussed.

Major Comments:

[Figure]

1. Downscaling: the authors' prior published BCSA Downscaling Method yields 3,000 precipitation realizations that are constrained to NLDAS-2 daily spatiotemporal statistical structure. It is not clear to me how this approach avoids acting like a low-pass stochastic filter for increasingly extreme temperatures, droughts, or floods. Specifically when contemplating more extreme quantiles that are rarely observed or have not been observed. The GCMs themselves are not strongly capable of capturing extremes. Moreover, limits in the observation record reduce the value for NLDAS-2 daily statistics in capturing extremes. Likewise, bias filtering also often eliminates extreme events. It is not clear to me how well the authors have even captured stationary extremes.

2. Human use scenarios: Although I understand that the authors are managing the computational demands of their work, the experiment being presented lacks a balance in how it treats humans versus climate in a manner that likely pre-ordains their attained results and ultimately may make them poorly representative of the uncertainties and impacts from the human decisions in the system. I found the human scenario justifications to be lacking in clarity and justification for their appropriateness. I suspect had the authors even done a basic parametric uncertainty for the aquifer conductivities that many of their claimed inferences would disappear into neglected parametric uncertainty effects. Moreover, the underlying "off/on" nature of the eight scenarios described in lines 237-271 mix mean behaviors and oddly unlikely human use combinations.

3. Global sensitivity analysis: the authors claim a variance decomposition is being done, but by merit of their experimental design the core potential for generating variance in the model is strongly concentrated within their climate sampling. Variance decomposition is strongly influence by factor ranges and deterministic human scenarios are extreme a priori statistical assumptions that strongly under sample the human component of the work. Additionally, the authors report only 1st order indices, which is tacit to a One-at-a-Time analysis in only highlighting separable single factor effects (e.g., Table 4 clearly indicates that a Total Order index in contrast to the 1st order index should be analyzed).

[Figure]

Minor Comments:

1. Introduction: at several points in the text (see lines 36-39; 49-53; 59-61; 75-80) the authors declaratively enumerate the existence of literature without any analysis for connection to this work and its novel contributions. Simple listing citations is not the same as providing readers with a guided narration of strengths, weaknesses, needs, and clarifying your own contribution.

2. At several points in the Methods it was not clear what was new or novel in this work relative to prior published work.

3. In terms of sensitivity analysis results, I would encourage the authors to improve their work by bootstrapping and reporting the confidence of their reported variance decomposition.

4. I found the figures poorly designed and difficult to interpret. Even Zooming to 200%, many of the claimed insights were not easily interpretable.

---

## Referee Comment (RC3) · Anonymous Referee #3 · 29 May 2018

Evaluation of impact of climate change and anthropogenic change on regional hydrology

S. Chang, W. Graham, J. Geurink, N. Wanakule, and T. Asefa

U of Florida and Tampa Bay Water

Abstract: General circulation models (GCMs) have been widely used to simulate current and future climate at the global scale. However, the development of frameworks to apply GCMs to assess potential climate change impacts on regional hydrologic systems and compliance with water resource regulations is more recent. It is important to predict potential impacts of future climate change on streamflows and groundwater levels to reduce risks and increase resilience in water resources management and planning. This study evaluated future streamflows and groundwater levels in the Tampa Bay region in west-central Florida using an ensemble of different GCMs, reference evapotranspiration (ET0) methods, and water use scenarios to drive an integrated hydrologic model (IHM). Eight GCMs were bias-corrected and downscaled using the Bias Correction and Stochastic Analog (BCSA) downscaling method and then used, together with three ET0 methods, to drive the IHM for eight different human water use scenarios. Results showed that changes in projected streamflow were most sensitive to GCM selection, however, projections of groundwater level change were sensitive to both GCM and water use scenario. Projected changes in streamflow and groundwater level were relatively insensitive to the ET0 methods evaluated in this study. Six of eight GCMs projected a decrease in streamflow and groundwater level in the future regardless of water use scenario or ET method. These results indicate a high probability of a reduction in future water supply in the Tampa Bay region if environmental regulations intended to protect current aquatic ecosystems do not adapt to the changing climate.

General comments:

The authors have presented an evaluation of the relative sensitivity of a water system in western Florida to a variety of forcings: precip/temp (via GCMs), evapotranspiration calculation method, and human water use scenario. They find that the system is relatively insensitive to ET calculation method, as well as to water use scenario. The authors conclude that the system is most sensitive to GCM projection. The quality of writing is good, and the results figures are professional. However, I have methodological concerns with the work, as well as concerns with the presentation of results. I am not sure that the concerns can be addressed in a straightforward re-write, but maybe they can. Most importantly, I think that the authors need to: 1) better justify the claim that RCP doesn't particularly matter; 2) use a physically-based ET calculation method (and not a temperature based method like Hargreaves); 3) present model calibration/validation for the hydrologic model and the groundwater model, as well as the human water system model; 4) show where the water use scenarios come from and why the authors feel justified in not changing land use. If the authors were able to do all of these things, I would re-review, but if they cannot, I think it would be best not to publish the work in HESS.

Specific comments:

Line 45: the authors note that the GCMs have biases that prevent accurate reproduction of historical hydrological conditions, but do not address those biases. The bias correction and downscaling methods mentioned do not correct for problems with the large-scale synoptic forcing that results in the failure of GCMs to reproduce natural variability (e.g., precipitation timing, variance, low frequency oscillatory behavior), and therefore are not particularly useful for use in driving hydrologic models. They are

especially poor at the precipitation extremes (flood and drought). I cannot agree that a climate change analysis should be driven with downscaled, bias-corrected GCM output.

Line 150: only RCP 8.5 was used because previous work showed choice of RCP to be less important than choice of GCM or ET estimation method. And yet you found choice of ET estimation method to be essentially unimportant here. I am suspicious of this claim. RCP 8.5 has very much more ET potential than does RCP 2.6. I would like to see it demonstrated that the difference between those two scenarios is insignificant for hydrology. That has not been my experience.

Line 107: you use HSPF and Modflow in something called IHM, but don't show calibration validation. Calibration/validation is essential for this work. How does the combined tool do with low flow versus high flow? What can you really know about groundwater contribution? A number of statistics are given in this paper (lines 155-160) about actual evapotranspiration over the historical period, but how is this really known? You know precip, and you know streamflow, but you don't know either groundwater infiltration or evapotranspiration, so you're just guessing at which portion is which, aren't you? I'd like to see your confidence in these numbers better justified.

Line 140: why is NLDAS-2 a good choice for bias-correction? What are the accuracy/biases of NLDAS-2?

Line 147: Your historical period is only 24 years. Are you confident that that is long enough to capture variability properly?

Line 200: Please provide calibration/validation results for the AFSIRS model. Is AFSIRS using Penman-Monteith for evapotranspiration? Can Hargreaves really substitute?

Line 243: what does item (7) in this list mean?

Line 247: why is irrigation assumed to be 85% efficient? That seems to me to be an important sensitivity.

Line 289: the only 2 equations presented are poorly described and confusing. Please put in terms of this study. It is not clear how the results are useful and interpretable. Is it a sensitivity in long-term average hydrology? A sensitivity in conditionality? What are the conditional relationships shown in (2)? Very difficult to make sense of how these relationships are applied in the results tables.

Line 331: Only Hargreaves was used. This is hugely problematic. Temperature-based evapotranspiration methods are empirical in nature, and have very high sensitivity to temperature that causes them to "overestimate ET in a way that is greatly at variance with the fundamental principle of conservation of energy at the land surface" (Lofgren and Rouhana (2016) "Physically Plausible Methods for Projecting Changes in Great Lakes Water Levels under Climate Change Scenarios", Journal of Hydrometeorolgy, 17, 2209-2223). You cannot perform a climate change assessment with pumped up temp values using a temp-only evapotranspiration calculation.

Line 474: I don't think the results "clearly show" this. Much hand-waving is done.

Figure 2: I'm not sure that boxplots are the best way to show this. There are trends that get obscured, aren't there?

Figure 3: Where are the historical baselines on these CDF's? How well does each GCM perform relative to the historical? Even after BCSD, probably big misses in retrospective relative to historical observed.

For sensitivity analysis, the ranges matter, don't they? So how were the ranges of change in pumping, ag, etc., determined? Local expert elicitation? Where do these projections/expectations come from?

Difficult to parse whether this paper si talking about changes in long term means or changes in variability. Where is the discussion of changes in variability/extremes? Hugely important for how much water will actually infiltrate versus evaporate. Changes in timing/duration/intensity/frequency of precipitation. And was surface storage modeled for its effect on evaporation and long-term infiltration? I didn't see that.

---

## Author Comment (AC1) · 29 Jun 2018

We appreciate the thoughtful comments from the reviewer, which have helped us to improve the original manuscript significantly. We explain in detail how we responded to the reviewer's comments, with line numbers referring to the revised manuscript unless otherwise noted.

We have attached the authors' response for reviewer's comments as well as revised manuscript as a supplemental file. We would really appreciate if you could check both files for further review.

Please also note the supplement to this comment:

https://www.hydrol-earth-syst-sci-discuss.net/hess-2018-91/hess-2018-91-AC1-supplement.zip

---

## Author Comment (AC2) · 29 Jun 2018

We appreciate the thoughtful comments from the reviewer, which have helped us to improve the original manuscript. We explain in detail how we responded to the reviewer's comments, with line numbers referring to the revised manuscript unless otherwise noted.

We have attached the authors' response for reviewer's comments as well as revised manuscript as a supplemental file. We would really appreciate if you could check both files for further review.

Please also note the supplement to this comment:

[Figure]

https://www.hydrol-earth-syst-sci-discuss.net/hess-2018-91/hess-2018-91-AC2-supplement.zip

---

## Author Response (AR1)

**<Author's response>**

Journal: HESS

Title: Evaluation of impact of climate change and anthropogenic change on regional hydrology.

Author(s): S. Chang et al.

MS No.: hess-2018-91

MS Type: Research article

**Editor**

We appreciate the thoughtful comments from the editor, which have helped us to improve the original manuscript significantly. We explain in detail how we responded to the reviewer's comments, with line numbers referring to the revised manuscript unless otherwise noted.

| Index |          | Comments                                                                                                                                                                                                                                               |  |  |  |  |  |
|-------|----------|--------------------------------------------------------------------------------------------------------------------------------------------------------------------------------------------------------------------------------------------------------|--|--|--|--|--|
| 1     | Editor   | The authors were provided with three detailed and substantive reviews, each of                                                                                                                                                                         |  |  |  |  |  |
|       | decision | which pointed out significant issues associated with the assumptions, methods and                                                                                                                                                                      |  |  |  |  |  |
|       |          | framing of results. While the authors have made a serious and good faith effort to                                                                                                                                                                     |  |  |  |  |  |
|       |          | address these issues, it is clear that the manuscript remains imperfect, at least with respect to its stated intent of identifying and quantifying the relative impacts of                                                                             |  |  |  |  |  |
|       |          | climate change and anthropogenic change on regional hydrology.                                                                                                                                                                                         |  |  |  |  |  |
|       |          | onetheless, the authors have made appropriate clarifications in several areas and
covided caveats that appropriately circumscribe their claims with respect to their
ciliate interpret their results may breadly and the paper does appear to be |  |  |  |  |  |
|       |          | ability to interpret their results more broadly, and the paper does appear to be
worthy of publication. My one request for a final revision would be that the                                                                                       |  |  |  |  |  |
|       |          | authors incorporate an abbreviated form of the final paragraph in Conclusions                                                                                                                                                                          |  |  |  |  |  |
|       |          | (which details study limitations) into their abstract so as to give potential readers a                                                                                                                                                                |  |  |  |  |  |
|       |          | clearer sense of the scope of the work.                                                                                                                                                                                                                |  |  |  |  |  |
|       | Author's | We revised the abstract of the paper to more clearly point out the new contribution                                                                                                                                                                    |  |  |  |  |  |
|       | response | and the limitation of this study.                                                                                                                                                                                                                      |  |  |  |  |  |
|       |          |                                                                                                                                                                                                                                                        |  |  |  |  |  |

[revised manuscript text omitted]

- 986